# Malaria parasite DNA-harbouring vesicles activate cytosolic immune sensors

Xavier Sisquella[1,2], Yifat Ofir-Birin[3], Matthew A. Pimentel[1,2], Lesley Cheng[4,5], Paula Abou Karam[3],
Natália G. Sampaio [1,2], Jocelyn Sietsma Penington [1], Dympna Connolly[6], Tal Giladi[3], Benjamin J. Scicluna[4,5],
Robyn A. Sharples[4,5], Andreea Waltmann[1,2], Dror Avni[7], Eli Schwartz[7,8], Louis Schofield[1,2,9], Ziv Porat[10],
Diana S. Hansen[1,2], Anthony T. Papenfuss [1,2], Emily M. Eriksson[1,2], Motti Gerlic [11], Andrew F. Hill [4,5],
Andrew G. Bowie[6] & Neta Regev-Rudzki[3]

STING is an innate immune cytosolic adaptor for DNA sensors that engage malaria parasite (*Plasmodium falciparum*) or other pathogen DNA. As *P. falciparum* infects red blood cells and not leukocytes, how parasite DNA reaches such host cytosolic DNA sensors in immune cells is unclear. Here we show that malaria parasites inside red blood cells can engage host cytosolic innate immune cell receptors from a distance by secreting extracellular vesicles (EV) containing parasitic small RNA and genomic DNA. Upon internalization of DNA-harboring EVs by human monocytes, *P. falciparum* DNA is released within the host cell cytosol, leading to STING-dependent DNA sensing. STING subsequently activates the kinase TBK1, which phosphorylates the transcription factor IRF3, causing IRF3 to translocate to the nucleus and induce STING-dependent gene expression. This DNA-sensing pathway may be an important decoy mechanism to promote *P. falciparum* virulence and thereby may affect future strategies to treat malaria.

[1] The Walter and Eliza Hall Institute of Medical Research, 1G Royal Parade, Parkville, VIC 3052, Australia. [2] Department of Medical Biology, The University of Melbourne, Grattan Street, Parkville, VIC 3010, Australia. [3] Department of Biomolecular Sciences, Weizmann Institute of Science, Rehovot 76100, Israel. [4] Department of Biochemistry and Molecular Biology, Bio21 Molecular Science and Biotechnology Institute, The University of Melbourne, Melbourne, VIC 3010, Australia. [5] Department of Biochemistry and Genetics, La Trobe Institute for Molecular Science, La Trobe University, Melbourne, 3086 VIC, Australia. [6] School of Biochemistry and Immunology, Trinity Biomedical Sciences Institute, Trinity College Dublin, Dublin 2, Dublin, Ireland. [7] The Institute of Geographic Medicine & Tropical Diseases and The laboratory for Tropical Diseases Research, Sheba Medical Center, Tel Hashomer 52621, Israel. [8] Faculty of Medicine, Sackler School of Medicine, Tel Aviv University, Tel Aviv 69978, Israel. [9] Australian Institute of Tropical Health and Medicine, James Cook University, Townsville, QLD 4811, Australia. [10] Flow Cytometry unit, Life Sciences Core Facilities, Weizmann Institute of Science, Rehovot 76100, Israel. [11] Department of Clinical Microbiology and Immunology, Sackler Faculty of Medicine, Tel Aviv University, Tel Aviv 69978, Israel. Xavier Sisquella and Yifat Ofir-Birin contributed equally to this work. Correspondence and requests for materials should be addressed to A.G.B. (email: agbowie@tcd.ie) or to N.R.-R. (email: neta.regev-rudzki@weizmann.ac.il)

Pathogens are sensed by pattern recognition receptors (PRR) of the mammalian innate immune system, which directly recognize pathogen-associated molecular patterns (PAMP), and also host-derived damage-associated molecular patterns (DAMP) that are released from infected host cells[1]. PRR activation is a double-edged sword; although it is the basis for the generation of an effective adaptive immune response, PRR activation can also drive pathology[2]. One PAMP recognized by PRRs is nucleic acid, and PRRs that detect pathogen nucleic acids are divided into two main groups. PRRs at the cell surface or in endosomes, such as Toll-like receptors (TLR), recognize pathogen-derived nucleic acids in the extracellular environment. By contrast, RIG-I-like receptors sense cytosolic viral RNA, and cGAS and IFI16, which signal via STING, sense cytosolic double-stranded DNA (dsDNA)[3]. Once activated, PRRs and dsDNA sensors trigger signaling cascades that alter gene expression and stimulate the production of type I interferons (IFN), chemokines and proinflammatory cytokines[4], which activates a broad anti-pathogen immune response.

Stimulation of cytosolic DNA sensors by pathogen dsDNA activates STING-dependent signaling to alter gene expression. When not active, STING is anchored as a homodimer to the endoplasmic reticulum (ER) membrane. Upon detecting pathogen dsDNA in the cytoplasm, STING becomes active and is able to bind TBK1, and together these proteins translocate from the ER, via the Golgi apparatus, to perinuclear endosomes. Once Ser366 is phosphorylated by TBK1, STING interacts with and activates IRF3. Phosphorylated IRF3 then dissociates from the STING–TBK1–IRF3 complex to form a homodimer and enter the nucleus to induce transcription of genes, including type I IFN genes (reviewed in ref. [5]).

However, parasites and other pathogens can target the same host sensors to promote their own survival[6, 7]. Even in the case of the intracellular malaria parasites that invade mammalian red blood cells (RBC), one study showed that parasite growth requires STING in immune cells[8]. That study suggested that *Plasmodium yoelii*, a rodent malaria parasite, needs STING to support its growth in laboratory mice[8], and it has also been shown that immune cell responses to genomic DNA from *P. yoelii*[8] and *P. falciparum*[9] are STING dependent. Thus STING-dependent sensing of malaria DNA is important to disease outcome, and may be considered either immune detection by the host, or immune escape by the parasite. However, the mechanism by which parasite genomic material is transferred from infected RBCs to the cytosol of host immune cells for STING-dependent sensing is unclear.

One mechanism pathogens utilize to transfer functional molecules into host cells is the secretion of extracellular vesicles (EV)[10]. EVs carry a multitude of proteins, lipids, metabolites and nucleic acids, which they transfer to target cells by way of fusion, providing secure and efficient delivery of cellular signals[11]. EVs are commonly classified according to their mode of biogenesis. Microvesicles (MV; 0.3–1 μm in diameter) are derived from the plasma membrane in response to external or internal stimuli by a process of evagination and vesicle formation. Exosomes are smaller in size (50–200 nm in diameter), formed inside multivesicular bodies, and are secreted independent of cell death[12]. Owing to their stability, EVs protect their cargo from degradation

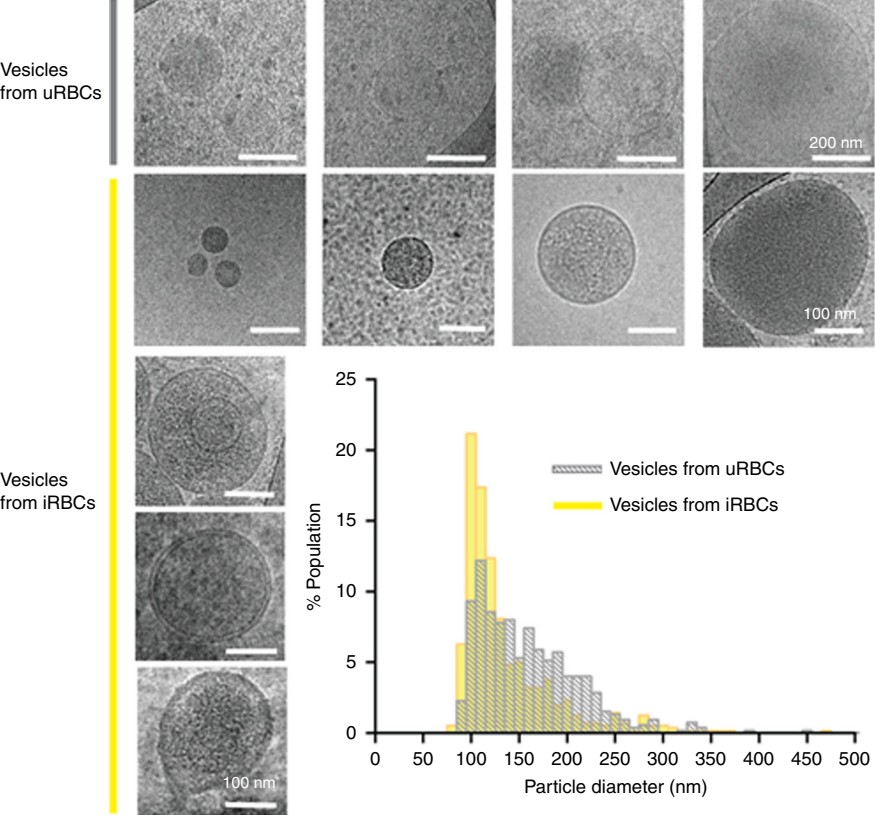

**Fig. 1** Characterization of vesicles released from *P. falciparum*-iRBCs. Cryo-EM images and size population data of vesicles from 3D7 and CS2 lines (bottom panels, scale bar 100 nm), uRBCs (top panels, scale bar 200 nm) and a small multivesicular population of EVs with a double membrane or containing one or two vesicles in the lumen (bottom left column, scale bar 100 nm). The qNano size distribution of EVs from iRBCs and vesicles from uRBCs (graph)

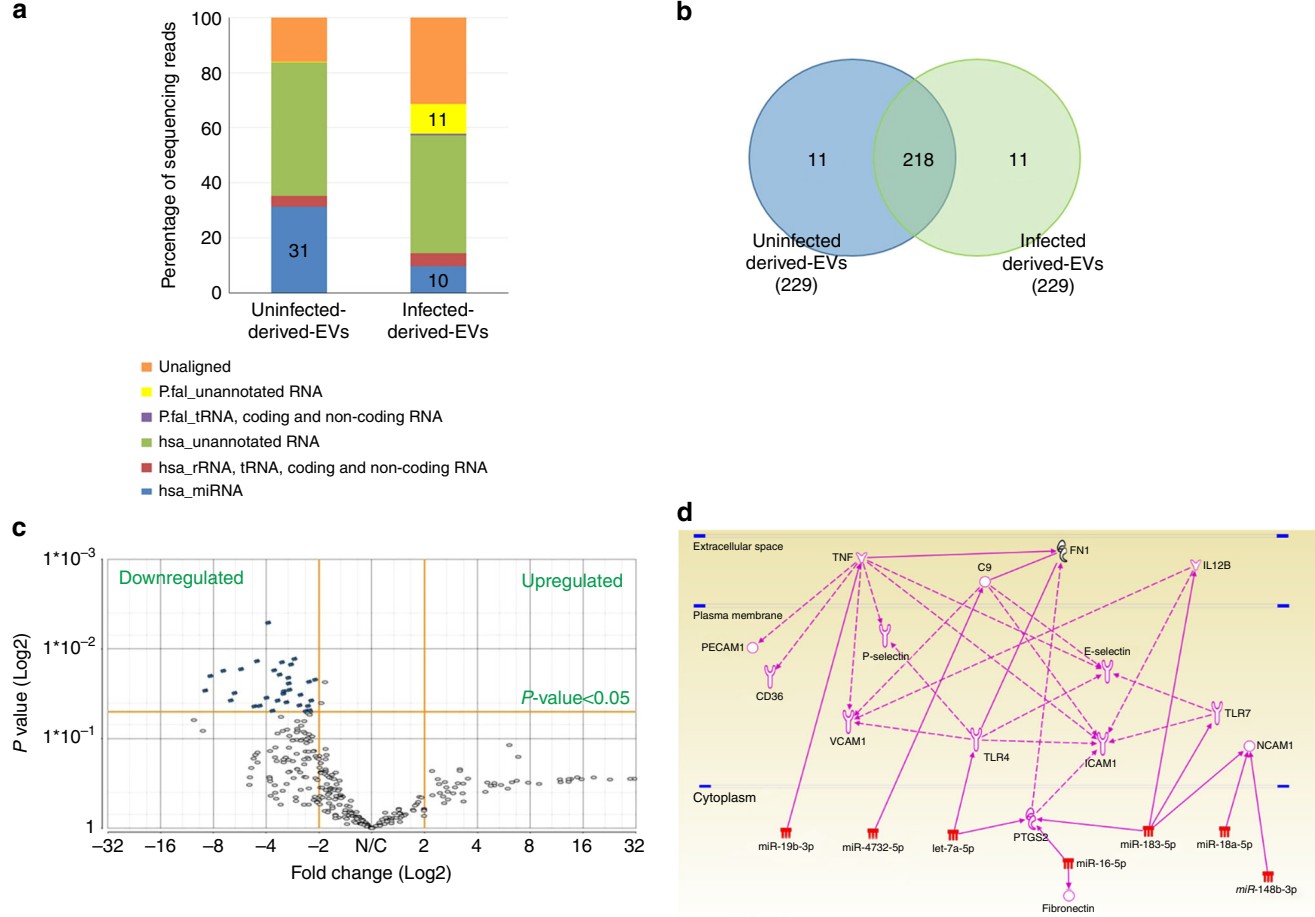

**Fig. 2** Packaging of the *P. falciparum* RNA species into EVs. **a** Bar graph of annotated and unannotated reads calculated as a percentage of total reads of each sample (Supplementary Table 1). Percentage alignment and mapping to *H. sapiens* (hg19; miRBase v.20, Ensembl v74 and tRNAscan-SE) and *P. falciparum* (ASM276v1; EnsemblProtists and tRNAscan-SE) genome and annotations from small RNA sequencing. **b** Venn diagram of the number of highly abundant miRNA species in iRBC-derived-EVs and uRBC vesicles. **c** Volcano plot of differentially expressed human miRNA in iRBC-derived EVs and uRBC-derived EVs (Supplementary Table 3). Normalized reads to RPM. ANOVA analysis, highlighted in blue: $p \leq 0.05$ (iRBC-ex vs uRBC-v) and $\pm 2.0$ fold change. **d** Pathway analysis of mRNA targets of miRNA differentially detected in EVs from iRBCs. Ingenuity Pathway Analysis[75] was used to find direct and indirect relationships, with the validated human mature miRNA differently expressed in EVs from iRBCs and mRNA related to malaria. All edges are supported by at least 1 reference from the literature or from canonical information stored in the Ingenuity Pathways Knowledge Base. Downregulation or the absence of these miRNAs in vesicles would allow protein translation to occur without interference

and denaturation in the extracellular environment. Although EVs from many cell types contain functional proteins[13, 14] and RNA[11, 15], the extent to which they also contain genomic DNA (gDNA) is less clear.

Cell-cell communication and sensing extracellular signals is essential for all living organisms, especially for pathogens[16], yet our understanding of these processes in parasitic protozoa is limited. This group of parasites is responsible for major diseases, including malaria, caused by the genus Plasmodium. *P. falciparum* and *P. vivax* cause most clinical cases of malaria. These parasites alternate between multiple developmental stages as they cycle between their mosquito and human hosts, encounter different environments[17]. Thus, these parasites require a range of efficient means to alter and even manipulate host responses. In the human host, blood-stage parasites cause the disease symptoms and pathology. The blood cycle is initiated when merozoites, the free and invasive blood-stage parasites, invade circulating RBCs. The 48 h asexual blood-stage cycle of *P. falciparum* involves differentiation of the invading merozoite through a ring stage, then trophozoite and schizont developmental forms[18].

*P. falciparum*-infected RBC (iRBC)-derived EVs have a function in parasite–parasite communication and promote the

differentiation of parasites in recipient cells to gametocytes[19, 20]. Malaria parasites also use EVs as a mechanism of intercellular communication to alter host cell responses[21, 22] using human RNA molecules[21, 23].

Individuals infected with *P. falciparum* or *P. vivax* have high circulating levels of platelet-derived and RBC-derived EVs[24, 25].

Here, we use nano-tools to study the nucleic acid cargo of parasite-derived EVs. We provide evidence that EVs released by *P.falciparum* parasites contain parasite non-coding RNA and parasite gDNA. The DNA is secreted via vesicles in a time-dependent manner, and is only detectable for the first 12 h after invasion of the RBCs. Our findings outline the process by which parasitic EV-DNA is transferred into the host cytosol, and detected by the STING-dependent cytosolic dsDNA sensing pathway to modulate host gene induction from a distance. We demonstrate that upon EV uptake, the downstream components of the STING-dependent pathway, namely TBK1 and IRF3, are phosphorylated, leading to the translocation of IRF3 into the nucleus to induce the transcription of host genes. Thus, the protected genomic DNA within the *P. falciparum*-derived EVs stimulates the STING–TBK1–IRF3 axis. Our data explain how malaria DNA elicits STING-dependent responses.

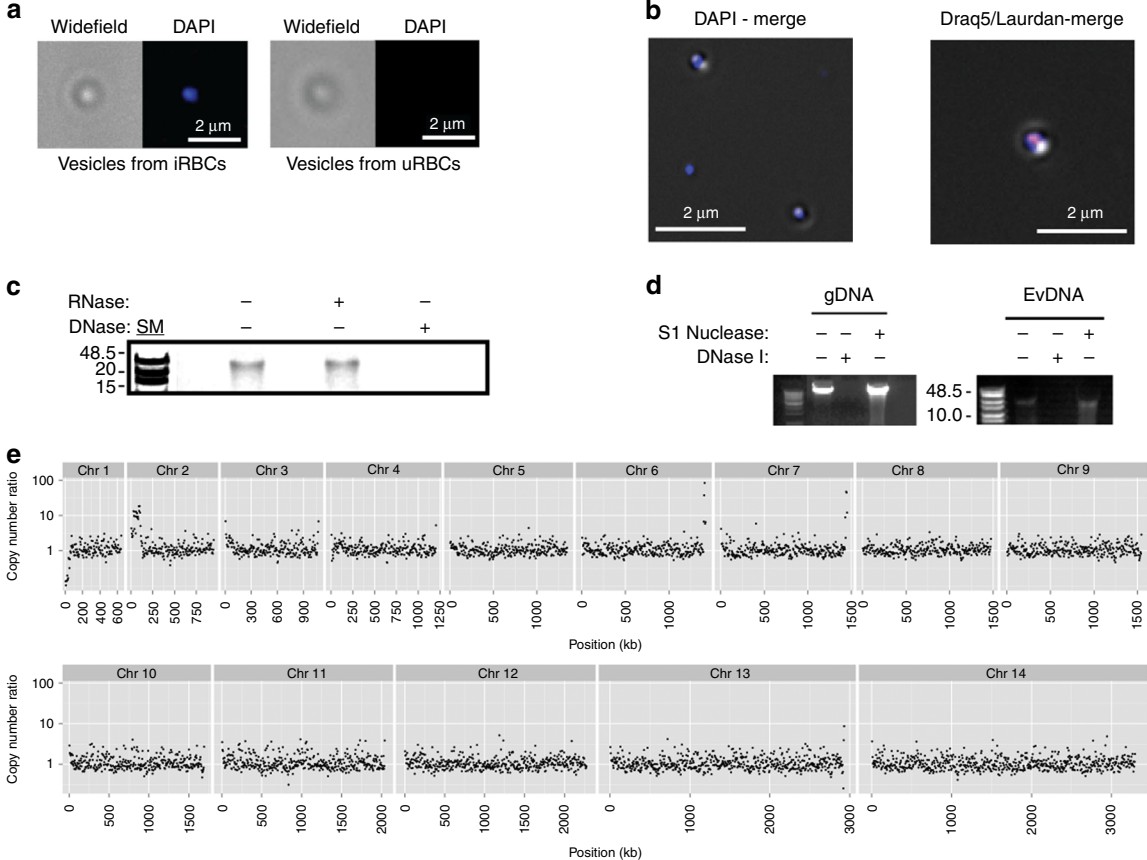

**Fig. 3** Parasite EVs released during the early post-invasion stage contain gDNA. **a** *P. falciparum*-derived EVs and EVs derived from uRBCs stained with DAPI and widefield merged. **b** Draq5-red and Laurdan-blue stained EVs (right). **c** Agarose gel electrophoresis of DNase/RNase-treated or DNase/RNase-untreated Ev-DNA. **d** DNA extracted from EVs was digested with either S1 nuclease or dsDNase I. Digestion of parasite gDNA serves as control. **e** Relative copy number fold change (RCNFC) between EV and gDNA control for nuclear chromosomes

## Results

### Characteristics of vesicles released from *P. falciparum*-iRBCs.
In previous work, we identified a cell-cell communication pathway mediated via 'exosome-like' nanovesicles released by ring-stage *P. falciparum*-iRBCs[20]. Here we further determine the biophysical characteristics of the secreted nanovesicles. Cryo-transmission electron microscopy (TEM) showed these vesicles to be composed primarily of unilamellar vesicles of 50–300 nm diameter in size; some possessed double or triple membranes (Fig. 1, bottom left panel), as was described previously[26]. Size and population analysis using nanopore-based particle detection (qNano), revealed a particle diameter distribution between 50 and 350 nm with a peak at $107.7 \pm 2.1$ nm, which corresponds to the typical size of mammalian exosomes (Fig. 1). qNano measured a broader diameter distribution with higher frequencies (125–225 nm) for vesicles purified from uninfected RBCs (uRBCs). Vesicles from uRBCs appear to be less electron-dense by cryo-TEM than those purified from infected cultures, suggesting the latter carry different cargo (Fig. 1, upper panel). The extracellular nanovesicles derived from ring-stage *P. falciparum*-iRBCs were further characterized by their buoyant density on an OptiPrep (OP) gradient and were found in fractions 2–5, between 1.03 and 1.07 g/cm³, which was similar to previous exosome reports[27]. Altogether, the morphology, size, number of membranes and density indicate that the nanovesicles released by *P. falciparum*-iRBCs are biophysically different to those of uRBC shedding vesicles and similar to those of mammalian exosomes. Moreover, *P. falciparum*-iRBCs secrete vesicles that originate from the cell internal milieu rather than by shedding of the host cell membrane.

### *P. falciparum*-iRBC EVs contain human and parasite small RNA.
It is well-known that EVs contain nucleic acid cargo[28, 29], therefore we analyzed the genetic cargo protected within *P. falciparum*-secreted EVs. Vesicles released by malaria parasites have been previously shown to contain human non-coding RNA molecules[29]. To determine whether EVs secreted from ring-stage iRBCs also contain parasitic small RNA, we isolated the entire repertoire of RNA found in *P. falciparum*-derived EVs and from uRBC vesicles used as a control. We then assessed the RNA content using a small RNA Bioanalyser assay (Supplementary Fig. 1). Our results showed that vesicles from uRBCs were found to contain little RNA (2.3–5.8 ng per sample) with no distinct peaks of tRNA or rRNA and low amounts of small RNA between 4–150 nucleotides. However, EVs secreted from *P. falciparum*-infected RBCs contained a significant amount of RNA ranging between 4 and 150 nt (10.2–47.3 ng per sample) with distinct peaks representing tRNA and potentially 5S RNA.

We performed small RNA deep sequencing to obtain a profile of the isolated RNA species. As expected, uRBC vesicles displayed a higher degree of alignment (83.7%) with the *H. sapiens* genome (hg19) as compared to the *P. falciparum* EVs (57.18% Supplementary Table 1). The RNA species of parasite EVs exhibited an 11.54% alignment with the *P. falciparum* genome where most are currently unannotated (Fig. 2a). However, the most abundant *P. falciparum* non-coding region was PF13TR011: ncRNA, found on chromosome 13, with the closest protein coding region being an uncharacterized protein (MAL13P1.461; Supplementary Table 2). These non-coding regions were not predicted to be potential precursors for mature microRNA

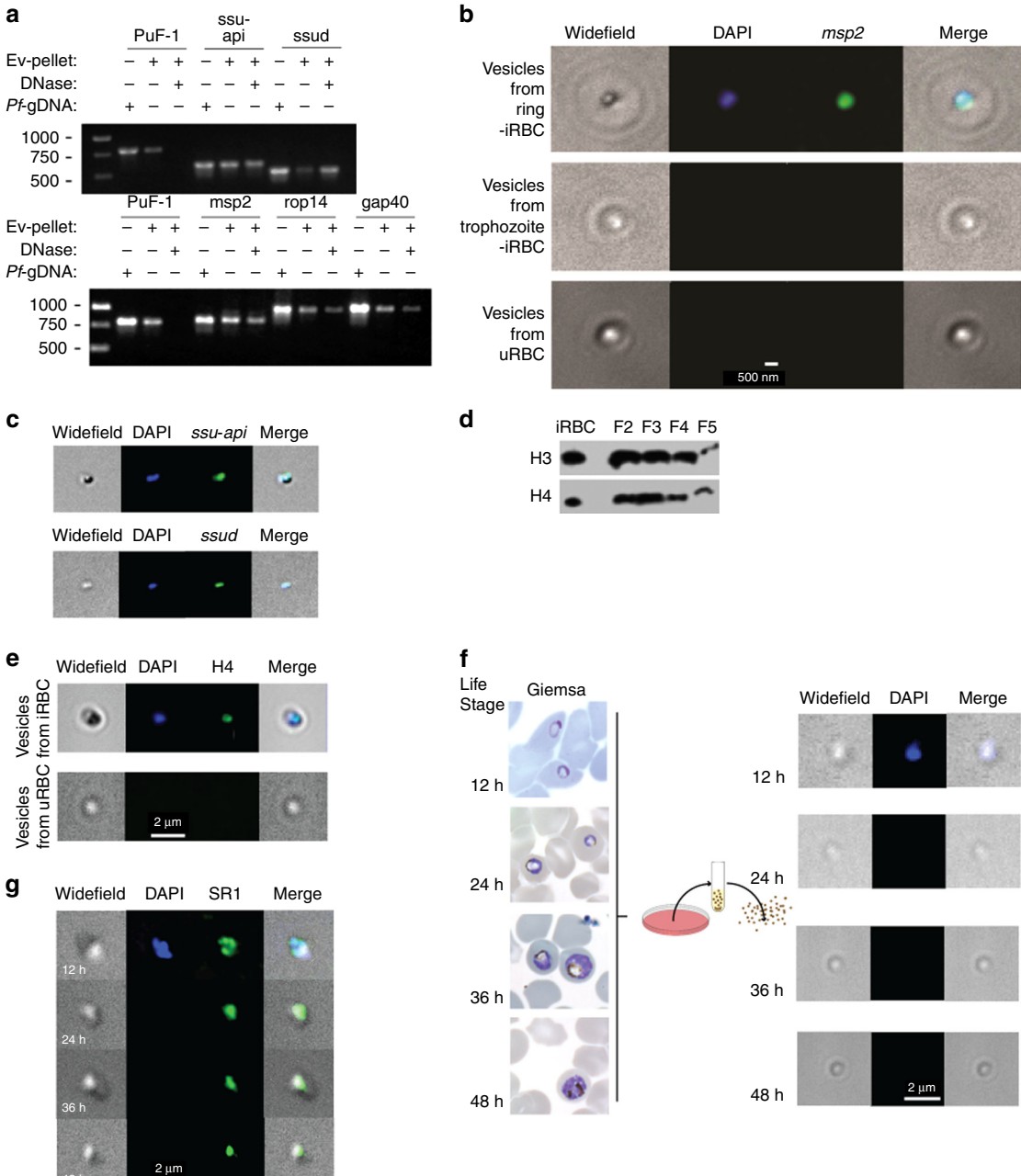

**Fig. 4** DNA-binding proteins and mitochondrial and apicoplast genes in EVs. **a** PCR for Ev-DNA markers, apicoplast (*ssu-api*), mitochondria (*ssud*) and nuclear genes (*msp2*, *rap14*, and *gap40*). Plasmid control PuF-1 was added externally to EVs prior to DNase I treatment. **b** *P. falciparum msp2* gene Ev-FISH images of ring and trophozoite EVs and uRBC vesicles. **c** *P. falciparum ssu-api* and *ssud* genes Ev-FISH of ring EVs. **d** H3 and H4 protein WB analysis for 2–5 OP gradient fractions (F). **e** *P. falciparum* H4 protein IFA. **f** *P. falciparum* parasites release EVs during the early post-invasion phase. Fluorescence microscopy using DAPI in EVs produced by iRBCs across their life cycle. Images of EVs collected at 12, 24, 36 and 48 h post invasion. Giemsa stains (first column) show the state of the parasites prior to collecting EVs at each time point. **g** SR1 control IFA

(miRNA) structures. Of note, to our knowledge, this is the first report of the identification of parasitic RNA delivered in EVs.

The percentage of reads obtained from mapping EV contents to human microRNA (miRNA) revealed uRBC vesicles contained a high percentage of miRNA reads (31%) compared to iRBC-derived EVs (10%). Out of the 229 human miRNA detected, 218 were detected in both uRBC-derived and iRBC-derived vesicles (Fig. 2b) with the most abundant of all being hsa-miR-451a.

Notably, the majority of the human miRNAs were present at much lower levels in the iRBC-derived EVs (Fig. 2c and Supplementary Table 3). Using quantative reverse transcription

PCR (qRT–PCR), we validated that the most significantly differentiated miRNA are present in iRBC-derived EVs at lower levels than in vesicles derived from uRBCs (Supplementary Fig. 2). Importantly, GO enrichment analysis revealed that the highest number of represented miRNA targets was of the functional group involved in cell adhesion regulation (Supplementary Table 4). Our Ingenuity Pathway Analysis led to the identification of seven miRNAs that influence various host cell adhesion genes, i.e., those encoding the Platelet endothelial cell adhesion molecule-1 (PECAM1), CD36, Vascular cell adhesion protein 1 (VCAM1), P-selectin, Intercellular Adhesion Molecule-

1 (ICAM1), E-selectin and Neural cell adhesion molecule (NCAM-1) host proteins (Fig. 2d). Many of these proteins are known receptors for PfEMP1-dependant iRBC adhesion factors. A case in point, VCAM1 is indirectly regulated by TNF[30], a predicted target of miR–19b–3p. These results support a study indicating that iRBC-derived vesicles carry functional human regulatory miRNAs to host endothelial cells and regulate VCAM1 expression in a yet unknown mechanism[23, 31].

In addition, we detected almost full mRNA coverage of the *P. falciparum* gene that encodes the early-transcribed membrane protein 11.2 (ID PF11_0040; Supplementary Fig. 3). This result was verified by targeted PCR, which detected the full-length mRNA (Supplementary Fig. 4), indicating the presence of intact parasite mRNA species within EVs.

**Ring-stage *P. falciparum*-iRBC EVs contain parasite gDNA**. We previously demonstrated that malaria parasites secrete vesicles that contain functional nucleic acid material (episomal plasmids)[20]. While the actual presence of gDNA in EVs is still largely undetermined, a few studies in mammals indicate that living cells release gDNA and even mitochondrial DNA protected within exosomes[32–34], yet the actual function of this gDNA remains unclear.

We used various approaches to investigate whether *P. falciparum* parasites deliver gDNA (i.e., endogenous genes) via EVs. EVs were collected from a wild-type NF54 strain, stained with the nucleic acid dye DAPI and imaged with Widefield Deconvolution Microscopy (WDM, Fig. 3a). A positive DAPI signal was obtained for iRBC-derived vesicles as compared to uRBC-derived vesicles. Additional staining with DNA stain Draq5 (red) and membrane dye Laurdan (blue) confirmed the co-localization of the DNA with the EVs, suggesting that the DNA was internal vesicle cargo (Fig. 3b). To ascertain that the DNA was indeed located within the vesicles, we extensively treated the EVs with DNase I prior to DNA extraction, as previously reported[33], and then employed the TapeStation DNA detection assay (Supplementary Fig. 5). This assay led to the detection of DNA species greater than 15 kb in size — large enough to encode multiple genes (Supplementary Fig. 5). Using differential enzymatic digestions, we confirmed these fragments to be DNA rather than RNA (Fig. 3c, Supplementary Fig. 11). Furthermore, treatment of EVs with DNases that recognize either single-stranded DNA (ssDNA) (S1 nuclease) or double-stranded DNA (DNase I) indicated that most of the DNA was dsDNA (Fig. 3d, Supplementary Fig. 11). To further characterize the Ev-DNA, we performed high-throughput whole-genome sequencing and comparative genomic hybridization analysis on DNase I-treated EVs. The sequence analysis revealed broadly distributed regions of the *P. falciparum* genome, representing sections from all its chromosomes (Fig. 3e, Supplementary Table 5). Mitochondrial DNA has been reported to be present in EVs[35, 36]. Of note, mitochondrial and apicoplast genomes were also present at lower copy numbers in the EVs compared to parental controls (Supplementary Fig. 6).

**Parasite genes are protected within released EVs**. We examined the nuclear, mitochondrial and apicoplast DNA EV content. To ascertain whether the different DNA species are protected inside the vesicles, we established a dedicated DNA protection assay. An external plasmid (PuF-1) was added to highly purified EVs followed by DNase I digestion and inactivation. While the external PuF-1 was digested in the presence of DNase I, the EV-DNA sequences of selected nuclear (*MSP2, ROP14, and GAP40*), apicoplast (*SSU-api*) and mitochondrial (*SSUD*) genes remained protected and detectable by PCR amplification (Fig. 4a,

Supplementary Fig. 11). This result indicates that *P. falciparum* genomic and organellar DNA are packaged into and protected within EVs.

To directly visualize the presence of specific DNA sequences in EVs, a fluorescence in situ hybridization (FISH)-based assay was developed (termed Ev-FISH). EVs were permeabilized with Equinatoxin II[37] and subjected to in situ DNA hybridization with nuclear, mitochondrial and apicoplast probes (*MSP2, SSUD, and SSU-api*, respectively) (Fig. 4b, c). Imaging allowed the direct detection of Ev-DNA content. While ring-stage-derived EVs were positive for DAPI, as well as *MSP2* (Fig. 4b, e), *SSUD* and *SSU-api* genes (Fig. 4c), no signal was detected for EVs collected from uRBC vesicles (Fig. 4e). To the best of our knowledge, this is the first report of a FISH assay performed on purified vesicles. For this reason we further assessed the validity of our assay by testing it on EVs from a parasite line expressing a GFP plasmid (3D7Emp3-GFP). Both blue and green signals were detected, corresponding to gDNA and the *GFP* gene, respectively, whereas only the signal for gDNA appeared in the parental 3D7 control (Supplementary Fig. 7), verifying the specificity of the assay. In conclusion, the molecular assays, imaging and DNA protection results (Figs. 3 and 4) confirm that *P. falciparum*-derived EVs indeed contain gDNA.

**Parasite DNA-binding proteins are present in EVs**. Western Blot analysis confirmed the presence of DNA-binding proteins, parasite histones H3 and H4, in EVs from the active OP density gradient fractions (Fig. 4d, Supplementary Fig. 11), suggesting that the DNA strands are compacted into nucleosome structural units within the EVs. Fluorescence immuno-labeling of EVs with antibodies against H4 (Fig. 4e) enabled detection of parasite histones within EVs and co-localization with gDNA (blue staining). gDNA also co-localized with known exosomal marker, SR1 (Sorcin)[38] (Fig. 4g).

**Parasite gDNA is delivered via EVs only at the ring-stage**. Only vesicles that are produced during the ring stage of the *P. falciparum* cycle were shown to be able to transfer episomal plasmids[20]. Thus, to determine whether the loading of parasite DNA cargo into EVs and its subsequent delivery occur through a selective process that reflects parasite biogenesis, we examined the presence of Ev-DNA in vesicles derived from highly synchronized parasite cultures collected at 12, 24, 36, and 48 h (h) after invasion into RBCs. gDNA was found in the EVs collected 12 h post invasion but was absent from EVs produced at later time points (Fig. 4f, b) or from gametocytes (not shown). We did, however, detect the SR1 positive exosomal protein control[38] at all time points (Fig. 4g). This finding indicates that DNA packaging and delivery via EVs happen only during the early stages following invasion, suggesting the existence of a time-dependent mechanism.

**Parasite EVs are efficiently internalized by monocytes**. Pathogens have developed an arsenal of strategies, including the use of EVs[16, 39], for evading and manipulating the immune response, in particular by directly targeting the sensors involved in nucleic acid immunity (reviewed in ref. [40]). Given the presence of *P. falciparum* genomic dsDNA in ring-stage-derived EVs, we examined the possibility that these parasites alter the innate immune response by transporting their own DNA species into the cytosol of immune cells and, thereby, stimulating dsDNA sensors. We stained *P. falciparum*-derived EVs for their DNA (using Hoechst DNA dye), RNA (using Thiazole Orange (TO) RNA dye) or lipid (using Dil lipid dye) cargo components, introduced them into monocytes (THP1 cell line) and monitored

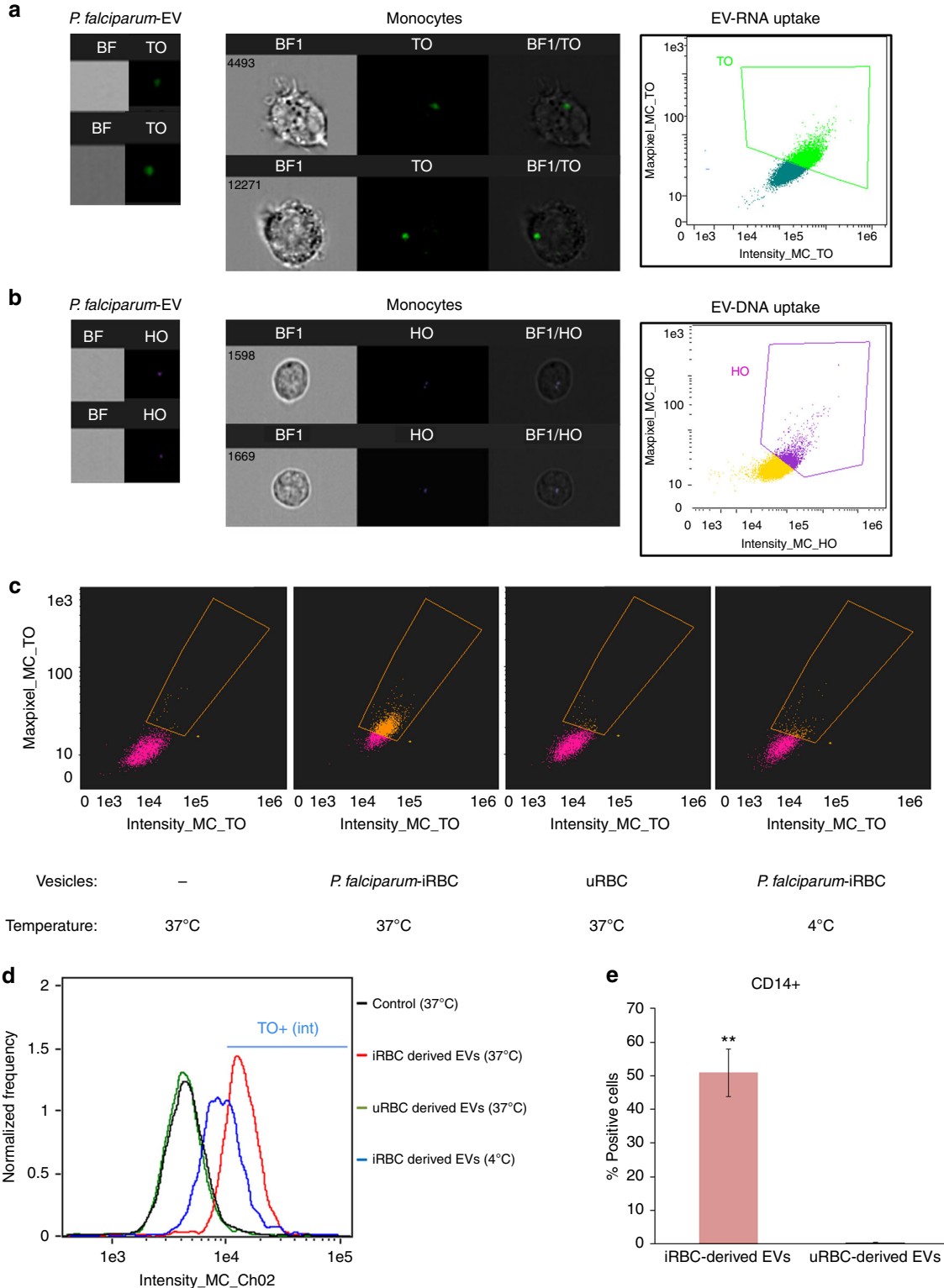

**Fig. 5** *P. falciparum* EV intake by monocytes. **a** THP-1 cells were incubated with RNA (TO)-labeled vesicles derived from *P. falciparum*-iRBCs and imaged by IFC. Graphs show (green) TO-labeled positive cells, gated according to unlabeled cells. **b** THP-1 cells were incubated with DNA (HO)-labeled vesicles derived from *P. falciparum*-iRBCs and imaged by IFC. Graphs show (purple) HO-labeled positive cells, gated according to unlabeled cells. **c** THP-1 cells were incubated with RNA (TO)-labeled vesicles derived from *P. falciparum*-iRBCs or uRBCs or THP-1 cells were incubated with RNA (TO)-labeled *P. falciparum* EVs at 4 °C and imaged by IFC. Graphs show (orange) TO-labeled positive cells, gated according to unlabeled cells. **d** Histograms of image stream analysis data in **c**. **e** PBMCs were obtained from nine different healthy donors and were incubated with TO-(RNA)-labeled *P. falciparum*-derived EVs; monocytes were stained for CD14 and imaged by IFC. Graphs present the percentage of TO-labeled positive cells, gated according to unlabeled cells from three repeats. SD and *T*-test analysis **$p \leq 0.001$. TO Thiazole Orange (for RNA staining), HO Hoechst (for DNA staining)

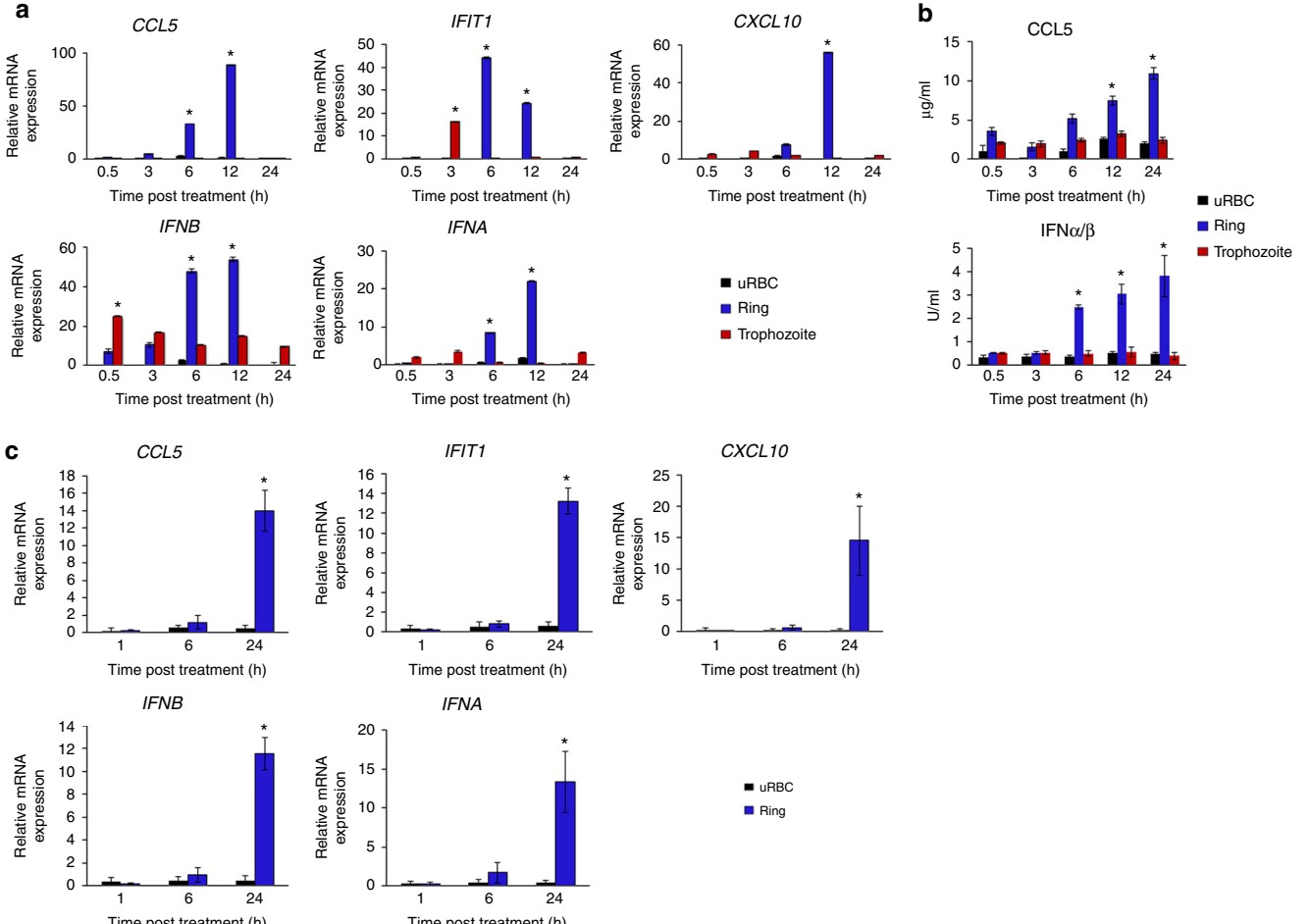

**Fig. 6** Ring stage *P. falciparum*-EVs activate immune gene induction in monocytes. **a** THP-1 cells were incubated with *P. falciparum* ring-stage, trophozoite-stage or uRBC-derived vesicles for 30 min, 3, 6, 12 and 24 h. RT–PCR was performed for the products of *IFNA*, *IFNB*, *CXCL10*, *IFIT1*, and *CCL5*. SD and *T*-test analysis *$p \leq 0.05$. **b** THP-1 cells were incubated with *P. falciparum* ring stage, trophozoite-stage or uRBC-derived vesicles for 30 min, 3, 6, 12 and 24 h. ELISA were performed for CCL5 and HEK blue IFNα/β. SD and *T*-test analysis *$p \leq 0.05$. **c** Human primary CD14+ cells were incubated with *P. falciparum* ring stage-derived or uRBC-derived vesicles for 1, 6, and 24 h. RT–PCR was performed for *IFNA*, *IFNB*, *CXCL10*, *IFIT1*, and *CCL5*. SD and *T*-test analysis *$p \leq 0.05$

their uptake using image flow cytometry (Fig. 5a, b and Supplementary Fig. 8). We were able to detect the EVs' nucleic acid (Fig. 5) and lipid (Supplementary Fig. 8) cargo within the cytosol of the recipient monocyte cells upon internalization of the vesicles.

Remarkably, the parasite-derived EVs exhibit a significantly higher level of intake by monocytes as compared to vesicles derived from uRBCs (Fig. 5c, d and Supplementary Fig. 8A). The internalization of the EVs, however, could be detected only under physiological condition at 37 °C (Fig. 5c, d). These results were further verified by tracking *P. falciparum*-EV intake by human monocytes extracted from PBMC (peripheral blood mononuclear cells), gained from nine healthy donors (Fig. 5e). Moreover, by performing live kinetic measurement, using IFC, we were able to monitor the dynamics of the DNA cargo distribution within monocytes (Supplementary Fig. 9). Labeled EVs were added to live THP1 cells and the derived signal was read continuously (after a 90–150 s loading time) by IFC for 45 min. As demonstrated, the transferred DNA signal intensity in the cells increased with time, indicating progressive uptake of the *P. falciparum*-labeled EVs by monocytes (Supplementary Fig. 9A, B). Within the first 15 min, the DNA signal appears as clear bright spots inside the cells, followed by a signal that progressively disperses through the cell (Supplementary Fig. 9C), suggesting that the DNA cargo's destination is the cytosolic milieu.

Together, these data indicate that vesicles secreted by the parasite exhibit a significant advantage to allow uptake by monocytes and are able to deliver their nucleic acid cargo into the cytosol of host immune cells.

**Parasite EV-DNA activates an innate immune response**. The fact that the destination of the EV-DNA cargo is the monocyte's cytosol provided a rationale as to how malaria DNA might gain access to DNA sensing pathways in immune cells such as monocytes, which is an unanswered question to date. To explore this possibility, monocytes (THP-1 cells) were treated with *P. falciparum* EVs derived from ring-stage (vesicles containing parasitic DNA; Fig. 6a blue bars) or trophozoite-stage (vesicles lacking parasitic DNA; Fig. 6a red bars). We then examined the ability of THP-1 cells to produce cytokines including IFNs that are typical of a DNA sensing response. Thus, at various points in time over the course of 24 h (30 min, 3, 6, 12 and 24 h) mRNA induction of *CCL5*, *CXCL10*, and IFNα (primers detect multiple isoforms of IFNα), *IFNB* and *IFIT1* (an IFN-stimulated gene) was measured (Fig. 6a). Notably, the mRNA levels of all five gene products were significantly induced upon intake of ring-stage *P. falciparum*-derived EVs (Fig. 6a, blue bars). At the same time, intake of trophozite-stage *P. falciparum*-derived EVs did not substantially increase mRNA induction from these genes (Fig. 6a,

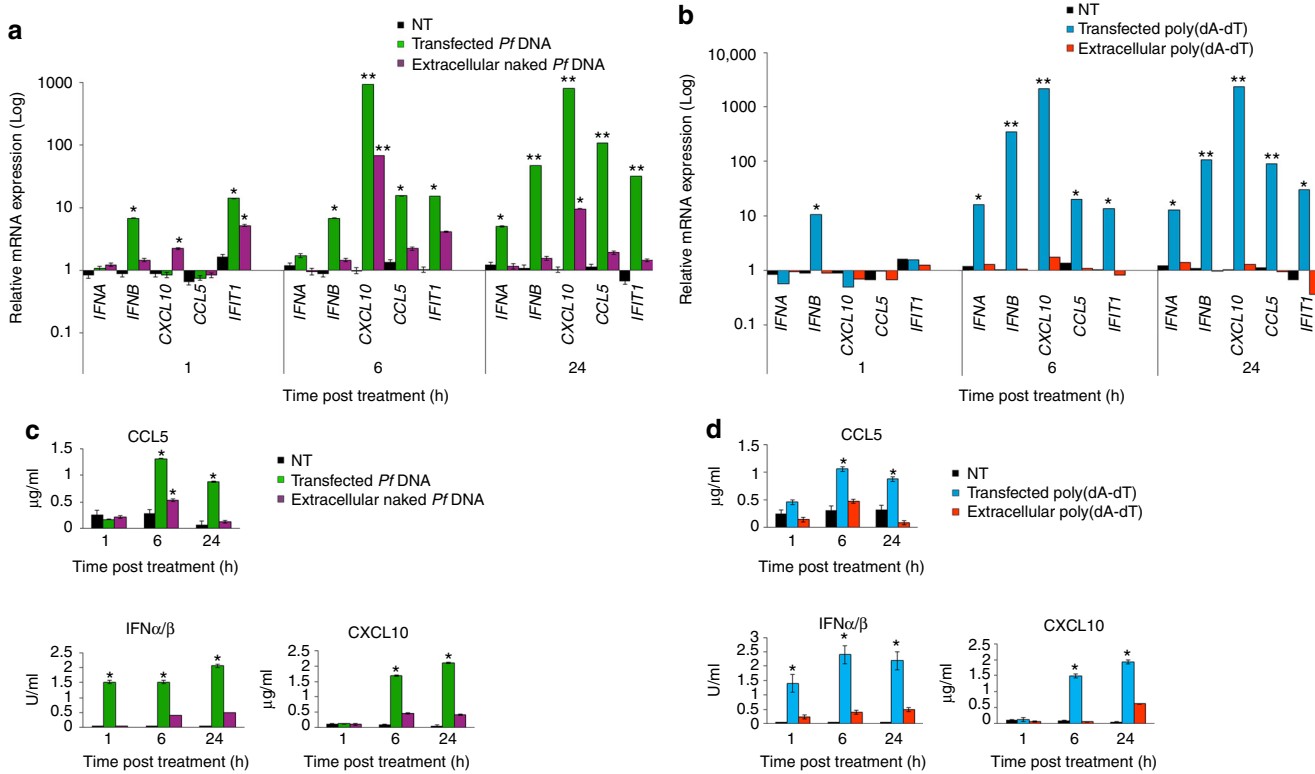

**Fig. 7** *P. falciparum* DNA stimulates innate immune gene induction in monocytes. **a** THP-1 cells were transfected or incubated with *P. falciparum* genomic DNA for 1, 6, and 24 h. RT–PCR was performed for the products *IFNα*, *IFNB*, *CXCL10*, *IFIT1* and *CCL5*. SD and *T*-test analysis *$p \leq 0.05$, **$p \leq 0.01$. **b** THP-1 cells were transfected or incubated with poly(dA:dT) for 1, 6, and 24 h. RT–PCR was performed for *IFNA*, *IFNB*, *CXCL10*, *IFIT1* and *CCL5*. SD and *T*-test analysis *$p \leq 0.05$, **$p \leq 0.01$. **c** THP-1 cells were transfected or incubated with *P. falciparum* genomic DNA for 1, 6, and 24 h. ELISA were performed for CXCL10 and CCL5 and HEK blue IFNα/β. SD and *T*-test analysis *$p \leq 0.05$. **d** THP-1 cells were transfected or incubated with poly(dA:dT) for 1, 6, and 24 h. ELISA were performed for CXCL10 and CCL5 and HEK blue IFNα/β. SD and *T*-test analysis *$p \leq 0.05$

red bars), nor were they affected by vesicles derived from uRBCs (Fig. 6a, black bars). These results indicate that the recipient monocytes sense the transferred *P. falciparum* EV-DNA within their cytosol and, as a consequence, activate their response. Our findings were also confirmed by the analysis of the secretion levels of RANTES (as measured by ELISA) and of type I IFN (as measured by bioassay) treated with vesicles derived from ring-stage iRBCs, trophozoite-stage iRBCs and uRBCs (Fig. 6b). Importantly, the induction of *CCL5*, *CXCL10*, IFNα, *IFNB*, and *IFIT1* mRNA was also confirmed using primary monocytes obtained from three healthy donors (Fig. 6c).

Next, we tested whether external parasite gDNA, rather than actively transferred gDNA, could be detected by monocytes and, as a result, stimulate a type I IFN response from the outside. We added naked *P. falciparum* gDNA (1 μg ml⁻¹) to the media of growing monocytes. After 24 h, cells were collected and the mRNA levels of *CCL5*, *CXCL10*, *IFNA*, *IFNB* and *IFIT1* were measured (Fig. 7a). As seen in Fig. 7a, much lower mRNA induction was detected in cells treated with naked *P. falciparum* DNA (Fig. 7a, purple bars) compared to the strong mRNA induction seen when monocytes were transfected with lipofectamine-containing *P. falciparum* gDNA (Fig. 7a, green bars). These results were confirmed by the secretion levels of *CCL5* and of type I IFN (Fig. 7c). The data for transfected *P. falciparum* gDNA exhibited a similar pattern of response to that observed following treatment with ring-stage *P. falciparum*-derived EVs (Fig. 6a, b). Furthermore, the observed immune response patterns to the insertion of *P. falciparum* DNA, either via transfection or EVs, is typical of a DNA sensing response in THP-1 cells[41, 42] and resembled the response of THP-1 cells to

poly(dA:dT) (Fig. 7b, d), which stimulates STING-dependent DNA sensing pathways in these cells.

Taken together, these data demonstrate that parasite-DNA-carrying vesicles can be internalized into monocytes to elicit an innate immune cytokine response.

**STING is required for EV-DNA-dependent gene induction.** Having established the effect of EV-DNA within monocytes, we wanted to establish whether the host response to the *P. falciparum* EV-DNA was indeed STING-dependent since it was previously reported that STING serves as an essential target for the development of malaria parasites[8], yet the mechanism by which the parasites activate the pathway by inserting its DNA into immune cells while growing within RBCs was not determined. Also naked gDNA (that may be released by parasite egress or lysis) added directly to cells led to a significantly lower cytokine response (Fig. 7), consistent with the requirement for intracellular cytosolic sensing. In order to determine whether STING has an essential role in the response for the recognition of EV-DNA, we used STING knockout (KO) THP-1 cells generated by the CRISPR/Cas9 system which were shown to be impaired in their ability to respond to cytosolic DNA, while their response to RNA was intact[43]. Upon *P. falciparum*-derived EV intake by the cells, mRNA induction responses were examined in a similar manner to that described above. Compellingly, although control THP-1 cells displayed robust induction of *CCL5*, IFNα, *IFNB* and *CXCL10* mRNA after *P. falciparum*-EV intake (Fig. 8a, blue bars), these responses were completely absent in STING KO cells (Fig. 8a, orange bars), similar to the responses to control vesicles

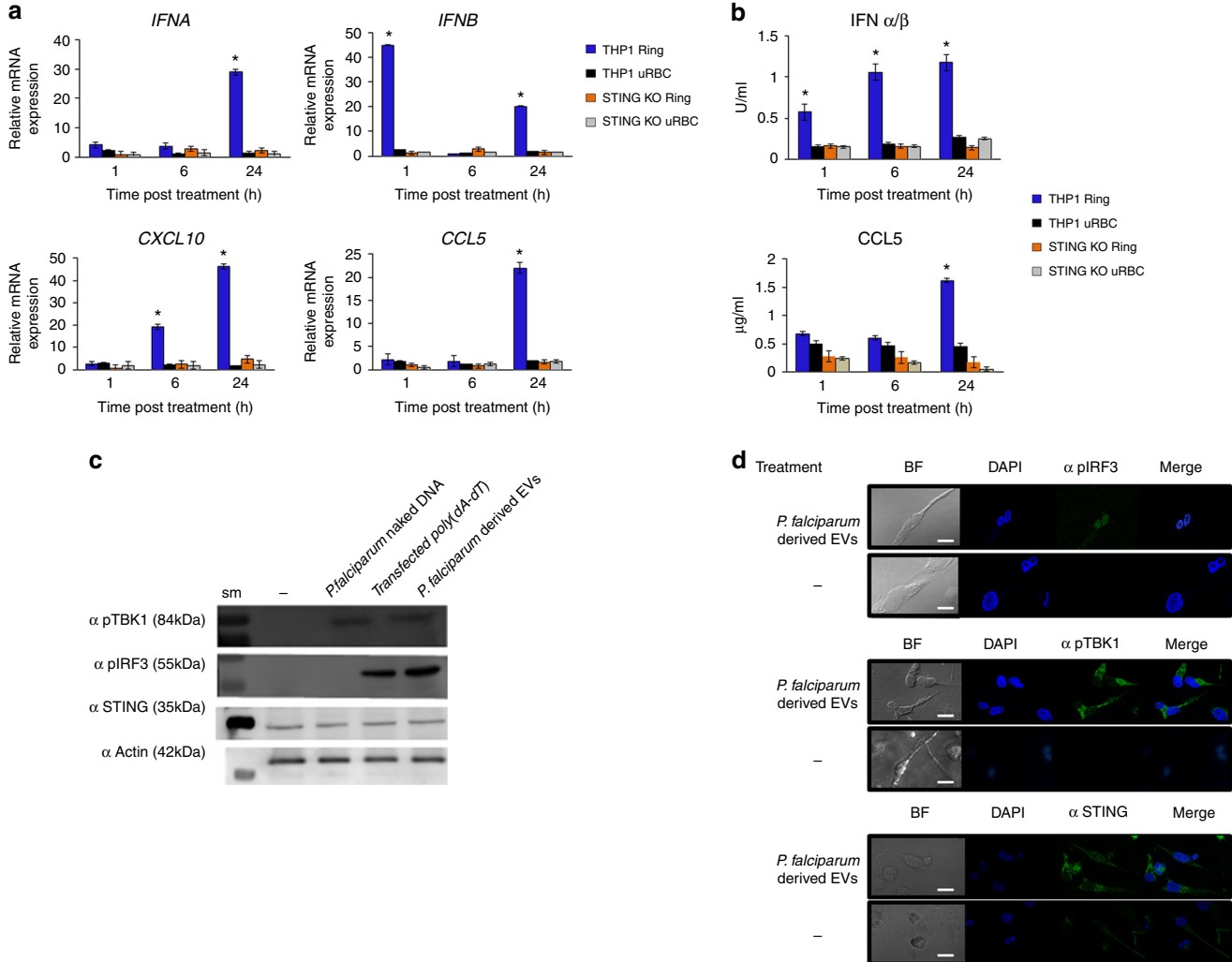

**Fig. 8** *P. falciparum* EV-DNA activates STING-dependent signaling in monocytes. **a**. THP-1 or STING KO THP-1 cells were incubated with *P. falciparum* ring-stage-derived or uRBC-derived vesicles for 1, 6 and 24 h. RT–PCR was performed for *IFNA, IFNB, CXCL10, IFIT1* and *CCL5*. SD and *T*-test analysis *$p \leq 0.05$. **b**. THP-1 or STING KO THP-1 cells were incubated with *P. falciparum* ring-stage or uRBC-derived vesicles for 1, 6 and 24 h. An ELISA assay was performed for CCL5 and CXCL10. HEK blue IFNα/β was performed. SD and *T*-test analysis *$p \leq 0.05$. **c**. THP-1 cells were incubated with *P. falciparum* ring-stage-derived vesicles, *P. falciparum* gDNA or transfected with poly(dA:dT) for 24 h. WB analysis was performed for STING, pIRF3, pTBK1 and α actin, sm-size marker. **d**. THP-1 cells were incubated with *P. falciparum* ring stage-derived vesicles for 24 h. Confocal microscopy images were taken for STING, pIRF3, pTBK1 (FITC), and DAPI. Scale bar 10 μm

collected from uRBCs (Fig. 8a, black bars). Similar results were obtained when *CCL5* or type I IFN production from control and STING KO THP-1s were measured (Fig. 8b, blue vs orange bars). Together, these findings indicate that STING plays a key role in type I IFN response to transferred *P. falciparum* DNA.

**TBK1 and IRF3 are activated after parasite-derived-EV intake**. The protein kinase TBK1 and the transcription factor IRF3 are essential downstream components of the STING-dependent type I IFN response. In unstimulated cells, both TBK1 and IRF3 are inactive in the cytoplasm. Upon stimulation by microbial dsDNA, however, cGAMP produced from cGAS binds STING, located on the endoplasmic reticulum membrane, leading to the recruitment of TBK1 to STING, and phosphorylation of TBK1 (pTBK1). pTBK1, in turn, then directly phosphorylates IRF3 (pIRF3), leading to the latter's translocation to the nucleus and stimulation of IFNβ induction[44, 45]. Thus, to further confirm that STING signaling is indeed activated by *P. falciparum*-EV uptake we examined whether TBK1 and IRF3 are activated and undergo

phosphorylation due to *P. falciparum*-EV internalization. For this, we utilized an antibody specific for phosphorylated TBK1 at Ser[172] (αp-TBK1) and an antibody against Ser[396]-IRF3. The treatment of THP-1 cells with *P. falciparum*-ring-derived EVs or transfected poly(dA:dT) as a positive control induced the phosphorylation of both TBK1 and IRF3 (Fig. 8c, Supplementary Fig. 11). In contrast, phosphorylation of TBK1 was not observed when naked *P. falciparum* dsDNA was added directly to monocytes (Fig. 8c, Supplementary Fig. 11).

As phosphorylated IRF3 dissociates from the STING-TBK1-IRF3 complex and enters the nucleus to induce the transcription of type I IFNs[44, 45] we monitored the subcellular localization of phosphorylated IRF3 following the internalization of *P. falciparum*-EVs by subcellular fractionation (Supplementary Figs. 10 and 11) and confocal microscopy (Fig. 8d). Indeed, as seen in Fig. 8d (upper panel), the phosphorylated IRF3 localizes to the nucleus upon treatment with *P. falciparum*-EVs and was detected in the nuclear fraction using WB analysis (Supplementary Figs. 10 and 11).

Together, these results demonstrate that malaria DNA-containing *P. falciparum*-EVs gain access to the cytosol of innate

immune cells and stimulate STING-TBK1-IRF3-dependent gene induction, providing for the first time a rationale as to how malaria DNA gains access to host DNA sensing pathways to modulate STING signaling.

## Discussion

The dynamic interplay between pathogens and their host is one of the most complicated themes in infectious disease progression. Pathogens excel in developing different means to facilitate cell-cell communication via, among other methods, secreted vesicles. EVs are vehicles that shuttle bioactive molecules between cells and they play an important role in many biological processes, while various pathogens utilize the exosomal pathway for their own benefits[16]. We have previously shown that *P. falciparum* parasites use vesicles to exchange active recombinant genes among them[20]. Here we provide a comprehensive analysis of the genetic material protected within vesicles released by wild-type *P. falciparum* strains, revealing that the malaria parasite delivers small non-coding RNAs and gDNA material that spans all its chromosomes, and also includes mitochondrial and apicoplast DNA. We further showed that the cytosolic immune machinery specifically senses the parasitic DNA via STING pathway.

The parasitic EV-DNA is released within the first 12 h post invasion, a stage in which the parasites do not yet replicate their DNA, suggesting that DNA packaging and delivery requires specific machinery. While the origin of the reserve DNA species during the early-stage of the parasite development is currently not clear (perhaps DNA remains that are carried by the invading merozoites), our findings are reminiscent of other exosome studies of mammalian cells demonstrating that the exosomal DNA represents the nuclear[32, 33] and mitochondrial[36] genomes. Currently, it is difficult to estimate the stoichiometry of DNA per EV, mainly as there is no reliable technique for separating the two types of vesicles in the samples those released from infected and from uninfected RBCs. Although further research needs to be pursued in order to understand how gDNA species are loaded into the EVs, the results of this study shed light on the previously reported finding that the ring stage in the parasite blood cycle is the critical stage for plasmid transfer between parasites[20]. For the parasite, export of gDNA protected within vesicles may also serve as a method for horizontal gene transfer[46, 47].

Innate immune sensing of both host and pathogen DNA is known to contribute to disease outcomes. Cells release DNA as a defense mechanism against danger, as does the microorganism Prochlorococcus, which releases its gDNA in vesicles[48] and even fragments of host gDNA[49] or mitochondrial DNA[50]. For malaria, responses of immune cells to genomic DNA from *P. yoelii*[8] and *P. falciparum*[9] were shown to be STING dependent.

Since malaria parasites face hostile environments in their complex life cycle, they must develop an ability to alter the host immune system. One line of reasoning suggests that innate sensing of malaria DNA by STING-dependent pathways favors the parasite and represents immune subversion, as is the case for other non-viral pathogens such as *Mycobacterium tuberculosis*, *Neisseria gonorrhoeae*, and *Staphylococcus aureus*[51–53]. *P. yoelii*, a rodent malaria parasite, requires STING to support its growth in laboratory mice[8], although the relevance of this to human malaria remains unclear. Malaria DNA has also been shown to be sensed by endosomal TLR9, which is expressed on pDCs, via a mechanism whereby malaria hemozoin targets the DNA to TLR9-positive endosomes[54]. Further, TLR9 has been shown to be essential for the development of protective immunity to *P. yoelii* in laboratory mice[55]. Thus for *P. yoelii* in laboratory mice, TLR9-dependent DNA detection is beneficial to the host, whereas STING-dependent DNA detection favors the parasite.

Whether STING-dependent sensing represents a pathogen decoy mechanism, and/or host detection of the pathogen requires further investigation for human malaria, but overall STING is likely to be important in defining disease outcomes. However, up to now it was unclear how malaria DNA gains access to cytosolic dsDNA sensing pathways that utilize STING in immune cells. Thus, we show that malaria DNA-containing *P. falciparum*-derived EVs gain access to the cytosol of monocytes and stimulate STING-TBK1-IRF3-dependent gene induction. Our finding provides a mechanistic explanation for host sensing of malaria DNA.

Interestingly, innate detection of pathogen nucleic acid by transfer into immune cells via exosomes was also shown for Epstein Barr virus[56]. There, cells latently infected with EBV triggered anti-viral immunity in neighboring DCs due to transfer of viral RNA via exosomes. Yu et al.[8] showed that removal of STING in vivo increases resistance of mice to *P. yoelii*, and that STING activation in pDCs downregulates TLR-dependent type I IFN responses via SOCS1. Thus it is possible that *P. falciparum* EVs if taken up by human pDCs would activate STING, which could amount to an immune subversion mechanism. However, as in the case of EBV, *P. falciparum* nucleic acid sensing at a distance by immune cells may represent a normal function of host immunity.

One of the major outcomes of malarial activation of nuclei acid sensing PRRs, including STING, is induction of type I IFNs. For example, Yu et al.[8] showed that early release of low levels of type I IFN favor host resistance to *P. yoelii*, while Wu et al.[57] determined that a strong type I IFN response to *P. yoelii*, that was dependent on RNA-sensing RIG-I-like receptors, contributed to a decline in parasitemia over time. Sharma et al.[9] showed that later release of high levels of type I IFN in response to *P. berghei* caused pathology and death, while another study showed that in *P. berghei*-infected mice, type I IFN signaling impaired Th1-dependent malaria immunity[58]. In contrast, a RIG-I-like receptor response to *P. berghei* RNA in infected mice was shown to be critical for host resistance to liver-stage plasmodium infection[59]. In humans, blood-stage *P. falciparum* infection induced type I IFN induction, leading to inhibition of innate and adaptive immune responses[60]. Thus, the relationship between type I IFN and protective vs pathological outcomes in malaria infection is complex and requires further investigation.

Apart from DNA, here, we demonstrate that the EVs released by *P. falciparum*-iRBCs contain not only human RNA species, as previously reported[31], but also small parasitic non-coding RNA molecules, similar to vesicles secreted by other pathogens[16, 28, 61]. Although we showed that the response of human monocytes to *P. falciparum* ring-stage vesicles, in terms of induction of type I IFNs and chemokines, was entirely STING-dependent (Fig. 7a, b), it is possible that host sensing of EV-RNAs also contributes to the immune response to *P. falciparum*. For example, EV-RNAs could be detected by TLR7 or TLR8 on certain host cells[62]. The presence of RNA species in *P. falciparum*-derived EVs provides a new perspective as to other potential means whereby the parasite can manipulate the host milieu. The miRNA and mRNA carried and delivered by secreted vesicles can control gene expression in recipient cells[29, 63]. We detected a significant population of *P. falciparum* non-coding RNAs in EVs, though the potential function of these molecules remains unknown. In addition, the predicted targets of the human miRNAs within the vesicles are genes involved in cell adhesion regulation, including VCAM1, CD36 and ICAM1 receptors, to which different forms of the virulence protein family PfEMP1 bind[64, 65]. These results strongly support the work published showing that *P. falciparum*-derived vesicles not only target endothelial cells and deliver functional miR-451a-Argonaute2 complexes (RISC) but that the RNA delivery results in elevation of a number of transcripts of the

VCAM1 receptor[23, 31]. Importantly, traffic of genetic material may serve as a sensing signal in a population to coordinate parasite social interactions[16], thus potentially playing a role in parasite differentiation into sexual forms as previously observed[19, 20].

In summary, we show that malaria parasites secrete EVs containing both parasitic RNA and DNA, and we demonstrate a novel mechanism by which parasitic DNA enters monocytes to modify the STING pathway response. Our comprehensive identification of the molecular cargo transferred by *P. falciparum*-derived EVs significantly enhances knowledge as to their potential in intercellular signaling and elicits key questions relating to parasite survival and malaria pathogenesis.

## Methods

**Cell culture**. The monocytic cell line THP-1[66], commonly used in innate DNA sensing studies, was used. The STING knockout THP-1 cell line was generated by the CRISPR/Cas9 system[43].

Parasite lines used were CS2eBsdGFP, 3D7edhfrGFP[20], NF54 (generously provided by Malaria Research Reference Reagent Resource Center (MR4)), CS2 and 3D7. *P. falciparum*: Parasites were grown in pooled donor RBCs provided by the Israeli blood bank (Magen David Adom blood donations in Israel) at 4% hematocrit, and incubated at 37 °C in gas mixture of 1% O₂, 5% CO₂ in N₂. Parasites were maintained in RPMI medium pH 7.4, 25 mg/ml HEPES, 50 µg/ml hypoxanthine, 2 mg/ml sodium bicarbonate, 20 µg/ml gentamycin and 0.5% AlbumaxII. *P. falciparum* cultures and human cell lines (THP1) were tested for mycoplasma once a month using commercial kits; MycoAlert Plus Kit (Lonza) or Mycoplasma Detection Kit—QuickTest (Tivanbiotech),

Primary monocyte isolation- Naïve PBMCs (obtained from Magen David Adom blood donations in Israel) were collected from three healthy donors. CD14 + cells were isolated from the PBMCs using CD14 MicroBeads and an LS column (MACS Miltenyi Biotech lnc). 1 × 10⁶ CD14+ cells were plated per well. The cells were incubated with *P. falciparum*-iRBC ring stage-derived vesicles or uRBC-derived vesicles for 1,6,24 h and were compared to the untreated CD14 + cells.

**Purification of extracellular vesicles**. Briefly, infected or uninfected RBC growth media was collected and cellular debris removed by centrifugation at 1500 r.p.m., 3000 r.p.m. and 10,000 r.p.m. The supernatant was concentrated using a Vivaflow 100,000 MWCO PES (Sartorius Stedium) and centrifuged at 150,000 × g to pellet nanovesicles. OptiPrep density gradient was performed as previously described[20]; OptiPrep velocity gradient purification—the initial vesicle pellet was resuspended in 2 ml incomplete media and further purified on a 10-ml continuous 10–30% Optiprep gradient made up in NTE buffer (137 mM NaCl, 1 mM EDTA and 10 mM Tris, pH 7.4), 250,000 g, 14 h, 4 °C, in a SW41 rotor (Beckman Coulter, Fullerton, CA, USA). Fractions (1 ml) were collected from the top.

**DNA isolation and analysis**. Nanovesicle pellets from 100 mL of culture were washed with PBS and incubated at 37 °C for 30 min with DNase I (Ambion). The enzyme was then inactivated (75 °C for 15 min), followed by acid-phenol:chloroform extraction.

**RNase and DNase digestion**. DNA was extracted using acid-phenol:chloroform as previously described[67]. EV-DNA or 3D7 gDNA were equally distributed and subjected to S1 nuclease, DNase or RNase A (Invitrogen) at 37 °C for 30 min. For the DNA protection assay, 25 pg of plasmid DNA PuF was added prior to DNase I treatment.

**Parasite EV time course**. CS2 parasites were grown to the late trophozoite/early schizont stage. The iRBCs were purified using a magnetized MACS column. Bound iRBC were eluted and added to fresh uRBCs. Culture was incubated on a shaker at 37 °C for 4 h to allow invasion. Remaining trophozoites/schizonts were lysed using 5% sorbitol. The culture containing early rings (0–4 h post invasion) at 5% parasitaemia and 4% hematocrit was washed twice with RPMI-HEPES and incubated at 37 °C. Culture supernatant was collected at 12 h intervals. After each collection, parasites were washed twice with RPMI-HEPES prior to addition of fresh medium. EVs were extracted from collected culture supernatant.

**Cryo-electron microscopy**. Cryo-TEM was performed as described previously[26] with a Tecnai G² F30 (FEI) transmission electron microscope operating at 300 kV with defocus between 10 and 16 µm across ×15,000 to ×39,000 magnification.

**RNA sequencing analysis**. RNA sequences were analyzed for quality control using FASTQC and the Torrent Suite 3.4.1. Output files (*.bam) were uploaded and aligned to the *H. sapiens* genome (hg19) using Bowtie 2, using Partek Flow (Partek

Incorporated, Singapore). Remaining unaligned reads were passed through for a second alignment with the *P. falciparum* 3D7 genome (ASM276v1) using Partek Flow. All aligned files were then analyzed using Partek Genomic Suite software (Partek Incorporated, Singapore). Sequences aligned to HG19 were mapped to miRBase_v.20, Ensembl v74 and tRNAscan-SE. Sequences aligned to ASM276v1 were mapped using annotations from EnsemblProtist (ASM276v1), INSDC Assembly GCA_000002765.1, non-coding and protein coding annotations) and tRNAscan-SE. Reads were normalized to reads-per-million, calculated as follows: Number of sequenced reads/total reads × 1,000,000 normalized using linear generalized modeling. Low-abundant miRNAs with less than 10 read counts across all samples were removed and highly abundant miRNAs further analyzed. Differential analysis was performed using the Partek Genomics Suite using the statistical functions. Selection of miRNAs was based upon ANOVA comparing infected vesicles with uninfected vesicles. Significant changes in miRNA expression were expressed in fold change (LOG2) and defined as (inf-ex vs uninf-v) $p \leq 0.05$ and $\pm 2.0$ fold change.

**Validation of EV-miRNA by RT–PCR**. Additional samples were collected for validation and RNA was isolated as mentioned above using the miRNeasy kit (Qiagen). RNA (1 ng) was reverse transcribed to cDNA (TaqMan MicroRNA Reverse Transcription Kit, Applied Biosystems, Australia) according to the manufacturer's protocol with a primer pool containing all miRNA assays (TaqMan microRNA assays, 5×, Applied Biosystems). Sixteen assays, which included 12 targets and 2 additional assays (hsa-miR-144–3p, hsa-miR-19b–3p, see Supplementary Table 6 for assay IDs), were pooled in order to serve as possible normalization controls. The cDNA samples were pre-amplified (TaqMan PreAmp Master Mix Kit, Applied Biosystems) and qRT–PCR (TaqMan Fast Advanced Master Mix, Applied Biosystems) was performed using individual miRNA assays (TaqMan microRNA assays, 20×, Applied Biosystems) and run on the ViiA 7 Real-Time PCR System (Life Technologies). Reverse transcription and pre-amplification no template controls using primer pools and individual assays were also prepared to ensure there was no background amplification of miRNA assays. Raw Ct data was uploaded to DataAssist (Applied Biosystems) to calculate delta delta Ct (ΔΔCt) using appropriate normalization methods[68] calculated by DataAssist by Applied Biosystems.

**Library preparation and genome sequencing**. EV-DNA was prepared and indexed for Illumina sequencing using the TruSeq® DNA LT Sample Prep Kit v2–Set A (Illumina, FC–121–2001). The library was quantified using the Agilent 2200 TapeStation. Indexed libraries were then prepared for paired end sequencing on a HiSeq instrument using the v3 100 cycle kit (Illumina).

**DNA sequencing analysis**. Sequence data quality was assessed using FastQC. A laboratory strain of *P. falciparum* 3D7 isolated from erythrocytes was used as control. Sequence reads from the EVs and control were aligned to the *P. falciparum* 3D7 reference genome (PlasmoDB v 10.0)[69] using Bowtie 2 (v2.1.0)[70] with options "--very-sensitive-local --minins 100 --maxins 800." PCR duplicates were removed using Picard MarkDuplicates (picard.sourceforge.net). Local read depth was calculated by counting aligned reads starting in contiguous 5 kb windows and was normalized by dividing by the total count for the genome. The relative fold change in copy number was estimated by the ratio of EV to control read depth, normalized for library size, in each window:

$$CN_i = \frac{n_{e,i} + 1}{\sum_i n_{e,i}} \bigg/ \frac{n_{c,i} + 1}{\sum_i n_{c,i}} \qquad (1)$$

where $CN_i$ is copy number ratio for bin $i$, $n_{e,i}$ is the number of EV reads starting in bin $i$, and $n_{c,i}$ is the number of control sample reads starting in bin $i$.

**EV RNA isolation and analysis**. Nanovesicle pellet from 100 ml was washed in PBS and total RNA was isolated using the miRNeasy Kit (Qiagen). RNA yield, composition and quality were analyzed by the Agilent 2100 Bioanalyser using the Small RNA Kit (Agilent). For RNA sequencing analysis and validation of miRNA by RT–PCR, see Supplementary Materials and Methods.

**Scanning ion occlusion sensing analysis**. Scanning ion occlusion sensing analysis was performed using the qNano system (Izon, Christchurch, New Zealand). Sample size distributions were calibrated by Izon Control Suite 2.2 using calibration particles of 100, 200, and 400 nm in size.

**Nanosight particle analysis**. Vesicle size distribution and concentration was performed using Nanoparticle Tracking Analysis (NTA) (Malvern Instruments, Nanosight NS300). Sample size distributions were calibrated in a liquid suspension by the analysis of Brownian motion via light scattering. Nanosight provides single particle size and concentration measurements.

**Western blot and confocal antibodies**. Histone antibodies: H3-CenPA (Alan Cowman laboratory, at Walter and Eliza Hall, Australia 1:1000), The anti-

centromere protein A (CenPA) (or anti-histone H3) antibody was generated by cloning the 5′ 193 nucleotides of the histone H3 gene (PF13_0185) into the pGEX4 T expression vector to produce a GST fusion protein in BL21 cells. Rabbits were immunized with the fusion protein and antibodies were purified by affinity chromatography[71]. Histone H4 (Abcam, ab16483, 1:1000). Anti-rabbit antibodies against STING pathway proteins; pIRF3 (CST, mAb #29047 1:250), pTBK1 (CST, mAb #5483 1:1000) and STING (CST, mAb #13647 1:1000) and Lamin (nuclear control, CST mAb #12255 1:1000) were obtained commercially (Cell Signaling). Antibodies against actin (AAS69609C1:1000), Anti-rabbit antibody against Tubulin (AAS12651C 1:1000) were obtained commercially (ANTYBODY VER-IFY). Full-length uncropped blots are presented in Supplementary Fig. 11.

**Immunofluorescence**. AlexaFluor488 secondary anti-mouse antibody, goat anti-human biotin conjugate antibody and streptavidin-Cy5 were purchased from Life Technologies and Abcam. Purified EVs were fixed by adapting standard methods[72] in the following way: Coverslips (Menzel-Glaeser) washed with water (2×), ethanol (1×), water (2×) and methanol (1×) were carefully flame dried. Volumes (10–40 μl) of EVs resuspended in PBS were applied to the coverslips and left to air dry before dipping them in cold methanol. Samples were blocked with filtered 3% BSA (Sigma-Aldrich) in PBS for 1 h. Primary antibodies in 3% BSA PBS were incubated for 1 h and washed five times for 5 min with 3% BSA PBS. Secondary antibodies, also in 3% BSA PBS, were incubated for 45 min and then washed five times for 5 min with 3% BSA PBS. Prior to imaging, coverslips were air dried and 5 μl of mounting medium with DAPI (Vectashield, Vector Laboratories) was applied before sealing them on microscope slides with nail polish. Sorcin-SR1was obtained commercially (Abcam).

**Widefield deconvolution microscopy**. Vesicle immuno or FISH labeled were imaged by WDM (ex-em wavelengths: DAPI 381/99–435/48, FITC 461/89–523/36, Alexa647 and Draq5 621/43–676/34) with a 100×/1.4 Oil immersion objective on a Deltavision Elite microscope (Applied Precision, USA). Z-stacks were captured with a 200 nm step interval and then deconvolved using SoftWorx (Applied Precision, GE Healthcare), which uses a constrained iterative deconvolution algorithm. Measurements undertaken on sub-resolution beads (~170 nm) on the DAPI channel (ex-em 381/99–435/48 nm) using WDM determined a resolution of 215 nm in xy and 500 nm in z dimension for this channel.

**EV fluorescent in situ hybridization**. The EV-FISH preparation was developed as a new protocol for gene in situ detection in EVs. We generated fluorescence-labeled probes (Fluorescein) for the gfp, msp2, SSUD and SSU-Api genes using the Label IT Nucleic Acid Labeling kit according to the instructions of the manufacturer (Mirus, MIR 3225). Briefly, probes were made by PCR from 3D7 WT gDNA for gfp, msp2, SSUD and SSU-Api using Expand High Fidelity PCR System (Roche, 3300226) with 0.3 M primers in a 50 μl reaction. The cycling parameters were as follows: 95 °C for 45 sec, 55 °C for 45 s, 64 °C for 1 min, 30 cycles. The labeling reaction, using the Label IT kit, was performed by mixing 40 μl of each PCR reaction (100 ng/μl) with 5 μl 10× Buffer and 5 μl Label IT Reagent (Mirus, MIR 3225). All other steps were performed according to the manufacturer's recommendations.

Parasite-derived EVs were permeabilized with equinatoxin II (EqtII) in 1X SSPE (150 mM NaCl, 10 mM NaH$_2$PO$_4$, 1 mM EDTA) in a final volume of 50 μl for 5 min at room temperature. Fluorescence-labeled PCR probes were denatured in a HS solution (50% formamide, 10% dextran sulfate, 2XSSPE, 250 μg Herring sperm DNA) at 95 °C in a final volume of 50 μl for 5 min. EVs were mixed with denatured probes at a 1:1 ratio and incubated at 95 °C for 2 min in a final volume of 100 μl. Hybridization was performed overnight at 37 °C on a coverslip sealed with a GeneFrame (ThermoScientific, AB-0576). Following three washes with 1XSSPE, the GeneFrames were removed and the coverslips were mounted onto slides in Vectashield with DAPI (Vector, H-1500) and analyzed by confocal microscopy using a DeltaVision OMX V4 Blaze 3D Structured Illumination Microscopy (3D-SIM) System (Applied Precision).

**Vesicle intake by THP-1 cells or PBMCs**. Vesicles were purified from *P. falciparum*-iRBCs or uRBCs culture and were stained using Thiazole Orange (TO; for RNA content), Hoechst (HO; for DNA content) or Dil (for lipid content). THP-1 cells or naïve PBMCs (obtained from blood donors of Magen David Adom in Israel) were stimulated with equal amounts of EVs, as were measured by Nanosight, (Nanoparticle Tracking Analysis (NTA) Malvern Instruments Ltd.). ~5 × 10$^6$ EVs per 1 × 10$^6$ cells for 5 min at 37 °C, 5% CO$_2$. Vesicle intake was detected by image stream flow cytometry (IFC).

**Image stream analysis**. Cells were imaged using a multispectral IFC (ImageStreamX mark II imaging flow-cytometer; Amnis Corp, Seattle, WA, Part of EMD Millipore). For direct vesicle intake measurements, vesicles were labeled and equal amounts were imaged. The ImageStreamX uses calibration beads 3μm in diameter for measuring focus and flow speed. To exclude these beads from the acquisition, objects were gated according to their area and intensity of the side scatter channel (Ch06) and the uniform bead population was easily identified and eliminated. For live kinetics, labeled EVs were added to THP1 cells and immediately acquired, for a period of 45 min (following a 90–150 s gap for sample introduction into the instrument). At least 5 × 10$^4$ cells were collected from each sample and data were analyzed using the manufacturer's image analysis software (IDEAS 6.2; Amnis Corp). Images were compensated for fluorescent dye overlap using single-stain controls. THP1 cells were gated for single cells, using the area and aspect ratio features, and for focused cells, using the Gradient RMS feature, as previously described[85]. Cropped cells were further eliminated by plotting the cell area of the bright field image against the Centroid X feature (the number of pixels in the horizontal axis from the left corner of the image to the center of the cell mask). Vesicle internalization was evaluated by using several features, including the intensity (the sum of the background-subtracted pixel values within the masked area of the image) and the max pixel (the largest value of the background-subtracted pixel).

**Real-time PCR**. Type I IFN, CCL5, CXCL10 and IFIT1 gene expression was determined by real-time PCR, using SYBR-fast green detection systems (Applied Biosystems). Expression levels were normalized to HTRP expression. The data are presented as the fold induction over untreated controls for each phenotype. Data represent the mean 9 SD from either biological replicates or technical replicates. The following PCR primers were used: *IFNB*: forward, 5′-CTGCATTACCT-GAAGGCCAAG-3′ and reverse, 5′-TTGAAGCAATTGTCCAGTCCC-3′; IFNα (to detect multiple isoforms): forward, 5′-TGAAGGACAGACATGACTTTGG-3′ and reverse, 5′-TCCTTTGTCCTGAAGAGATTGA-3′; *CCL5*: forward, 5′-TCATTGC-TACTGCCCCTCTGC-3′ and reverse, 5′-TCCTTGACCTGTGGACGACT-3′ *CXCL10*: forward, 5′-GGCAATCAAGGAGTACCTCTCT-3′ and reverse, 5′-GCAATGATCTCAACACGTGGAC-3′; *IFIT1*: forward, 5′-CAC-CATTGGCTGCTGTTTAGCTCC-3′ and reverse, 5′-GGCAGCCGTTCTG-CAGGGTTT-3′; *HRPT*: forward, 5′-CCTGGCGTCGTGATTAGTGAT-3′ and reverse, 5′-AGACGTTCAGTCCTGTCCATA-3′

**Confocal microscopy**. For the visualization of STING, pIRF3 and pTBK1: Following incubation with *P. falciparum*-derived EVs, cells were fixed and permeabilized with 4% PFA and 2% Sucrose at 4 °C and were labeled with antibodies against STING, pIRF3 or pTBK1. Images were acquired on a Zeiss LSM 710 confocal microscope, using a 60× oil objective. Image processing was performed using Zen 2010 (Zeiss) and ImageJ. All images are representative of at least three independent experiments.

**Measurement of cytokine production from monocytes**. CXCL10 and CCL5 were measured by ELISA according to manufacturer's instructions (R&D Systems)[73], while type I IFN production was measured by bioassay, using HEK Blue IFN-α/β reporter cells (InvivoGen).

**Cytosolic and nuclear fractionation**. Preparation of cytosolic and nuclear fractions were performed as previously described[74].

Following treatment by *Pf*-EVs, cells were washed three times with ice-cold PBS, resuspended in three volumes of cold cell disruption buffer (KCl 10 mM, MgCl$_2$ 1.5 mM, Tris-Cl 20 mM and DTT 1 Mm) and incubate for 10 min on ice. Cells were transferred into a baked Dounce homogenizer and were homogenized using ~30 strokes with a baked, type B pestle. The homogenate was transfer into a fresh tube and treated with 0.1% Triton X-100. Cell nuclear fraction was isolated by centrifuging the homogenate at 1500 g for 5 min, the supernatant (containing the cytoplasmic fraction) was transferred into a new tube containing RIPA buffer (NaCl 10 mM, EDTA pH 8.0 5 mM, Tris, pH 8.0 50 mM, NP-40 1.0%, sodium deoxycholate 0.5% and SDS 0.1%) and protease inhibitor (Sigma-Aldrich, P8340). The nuclear fraction (pellet) was washed three times with PBS and resuspended in RIPA containing protease inhibitor. Both fractions were further analyzed by WB.

**Data availability**. Sequence data that support the findings of this study have been deposited in the European Nucleotide Archive with the primary accession codes PRJEB6718 and PRJEB6714. Other data that support the findings of this study are available from the corresponding author upon request.

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

## Acknowledgements

We thank Malaria Research Reference Reagent Resource Center (MR4) for their generous supply of parasite strains. Special thanks to Prof. Alan Cowman for his contribution to this study and for the scientific environment of his laboratory. We thank Dr Ken Pang, Dr Jacob Baum, Dr Maya Olshina and Ms Natalie Page for scientific discussions, Dr Bradley Coleman (Andrew Hill's lab) for the vesicle assay, Dr Kelly Rogers and Lachlan Whitehead (WEHI imaging facility), Stephen Wilcox (Systems Biology and Personalised Medicine), Dr Eric Hansen (the Advanced Microscopy Facility, Bio21 and the Melbourne Materials Institute (MMI), Mr. Vladimir Kiss (Department of Biomolecular Sciences, Weizmann Institute). M.G. was supported by the Israel Science Foundation (ISF) (grant #1416/15), alpha-1 foundation, Recanati Foundation (TAU), and Individual research grant Varda and Boaz Dotan Research Center. The research of Dr Neta Regev-Rudzki is supported by the Israel Science Foundation (ISF) (619/16 and 119034) and by the Benoziyo Endowment Fund for the Advancement of Science, the Jeanne and Joseph Nissim Foundation for Life Sciences Research and the Samuel M. Soref and Helene K. Soref Foundation. Dr Neta Regev-Rudzki is the incumbent of the Enid Barden and Aaron J. Jade President's Development Chair for New Scientists in Memory of Cantor John Y. Jade. A.B. and D.C. were supported by grants from the National Institutes of Health (AI093752) and Science Foundation Ireland (11/PI/1056).

## Author contributions

N.R.-R. and A.B. designed the experiments and wrote the paper. X.S., M.A.P., and N.R.-R. established EV purification assay, X.S. performed EV imaging and EV-DNA characterization in vivo. Y.O. co-led and performed immune host experimental work. M.A.P. performed EV-DNA in vitro characterization, DNase protection assay and extracting EV-DNA for sequencing analysis. L.C., N.R.-R., B.J.S., R.A.S., and A.F.H. performed EV RNA characterization and analysis. J.S.P. and A.T.P. performed EV-DNA sequencing analysis, A.W. helped in setting RNA qPCR characterization, P.A.K., T.G., and Z.P. assisted in image stream flow cytometry method and analysis for EVs, N.G.S., E. M.E., L.S., D.S.H., D.C., and M.G. advised on immune assays, D.A. and E.S. assisted in generating primary human monocyte experimental work.
