## [Peer review file · Nature Communications]

Reviewers' comments:

Reviewer #1 (Remarks to the Author):

Recent studies demonstrated that the exchange of exosomes and microvesicles secreted by Red blood cells (RBCs) that were infected with *P. falciparum* seem to trigger sexual differentiation, leading to the appearance of gametocytes in parasite cultures. Notably, the rate of formation of these parasite-modified EVs and the rate of gametocyte formation increased significantly in the presence of drug pressure. It was suggested that this novel mode of intercellular communication provides the parasite a route for sensing and escaping a hostile environment. There is much to say about this new communication methods across kingdoms and in particular many viruses co-opt the EV cell-cell communication pathways of their hosts.

It was shown that EVs released from Pf infected RBCs contain parasite Proteins and DNA. In their prior work the authors showed that *P. falciparum*-infected RBCs are capable of transferring DNA within the population via EVs that had exosomes like characteristics. When EVs derived from the infected RBCs are used to stimulate peripheral blood mononuclear cells (PBMCs) they were shown to stimulate monocytes (CD14+) and induce dose-dependent production of cytokines. Altogether, these prior results suggest that EVs released from Pf infected RBCs play a role in antigen presentation and the modulation of host immune responses in malaria.

In the current paper the authors claim that exosomes released by RBC infected with Pf parasites contain parasite non-coding RNA and also parasite gDNA. Interestingly the DNA is secreted via EV/exosomes in a time-dependent (staged) manner, and only detectable at the first 12 hours post invasion the RBCs. The authors further suggest that the parasite requires STING activation to sustain parasite growth and cite prior work (Yu et al., Immunity 2016).

The work here by is certainly suited for publication in nature communications as it describes an important and critical advance over current knowledge in the field of pathogen-educted EVs and thier biological fuction(s). However i have some major and minor issues that need addressing or shoudl be discussed.

major

- 1) The claim that the EVs containing pf-DNA are exosomes is a difficult one to make, and I'm not sure if this is crucial. The transfer itself of functional material is more important. If the authors wish to claim the Evs from uninfected |RBCs are 'vesicles' and the EVs from infected RBC are 'exosomes' in the sense that they are derived from MVBs (not by protein expression and physical characteristics), then interference of the MVB-exosomes pathway, i.e. GW-treatment, siRNA against snMase or ESCRTs etc. will be necessary.
- 2) The authors frequently use the term 'infected exosomes', this is confusing as they are not infectious. While the fish data of DNA in the exosomes is compelling, to understand what the authors are measuring in the bulk EVs with their omics approaches, any idea of the stoichiometry of DNA per EV/exosomes would be of relevance. Would it be possible to provide such information, for example by limiting dilution of the EVs and measuring DNA copy nrs by PCR? Alternatively, the authors could measure the concentration of EVs in their preps with particle analysis (nanosight and or TRPS).
- 3) The finding that DNA packaging and delivery via exosomes happen only during the early stages following invasion, suggests the existence of a time-dependent mechanism but would there no such differences to be expected at the m(i)RNA level?
- 4) To increase physiological relevance; would plasma EVs from infected individuals be expected to harbor parasite DNA and be useful as a diagnostic?
- 5) (minor point) The FACS data should also be presented as histograms are there some monocytes that engulf large amounts of EVs?
- 6) The authors claim that the Pf EVs are 'preferentially' internalized by CD14+ peripheral monocytes and THP1 cells. However I cannot see any data from CD14- cell types. In addition the

internalization is unclear.

7) It is unlikely that the RNAs measured and transferred via the Pf EVs/exosomes have no effect at all. Can the authors really exclude other mechanisms of IFN-induction via STING? For example the authors may use a TLR mimic (poly-IC) in the STING KO cells to show that the endosomal TLR3/7/8 sensing pathway is not compromised. Is there a possibility that exoDNA is transcribed into inflammatory RNAs in the recipient cells (Chiu et al., Cell 2007)? Also cGAMP is known to be incorporated into particles and is exchanged between cells (Bridgeman et al., Science 2015), are the PF-exosomes devoid of cGAMP?

Minor

1) Exosomes are typically 100nm in size, and the diffraction limit of light is 200nm. This suggests that it is impossible to tell whether the images in Fig 3A are indeed DNA-containing exosomes.

2) The statement that deep-sequencing leads to an 'unbiased' representation is given too much credit into the procedure. It is well known that every sequencing protocol and informatics analysis leads to some sort of bias.

3) I find the figures in general not very appealing, this needs more attention. There seems to be a higher degree of RNA-transfer than DNA (figure 5) is this true? If so is this not relevant to mention?

4) Yu et al. identify a cross-regulatory mechanism of two type I IFN signaling pathways in plasmacytoid DCs (pDC), which is critical for generating protective immunity against lethal malaria infection. They show that cGAS-, STING-, MAVS-, or MDA5-deficient mice were resistant to lethal YM infection compared with wild-type (WT) mice. This suggests that these molecules are required for pathogen survival and infection. However in this paper it was also shown that serum amounts of IFN- α /b in the genetically deficient mice were abolished upon depletion of pDC or ablation of Tlr7, Myd88, and Irf7. Interestingly, EVs from pathogen-infected cells that are taken up by pDCs recognize RNA (Baglio et al., PNAS 2016) may also protect against (viral) infection. Can the authors reconcile these observations with their findings? Would it be possible that Pf EVs have an alternate function in pDCs?

5) Figure 7 How did the authors control for equal amount of EVs (exosomes from uRBC vs EVs from iRBCs).

6) The fact that the EV-DNA must reach the monocyte cytosol upon uptake, suggests an endosomal/lysosomal escape mechanism. Would there be a reason to believe that Pf-exosomes have such a mechanism?

Reviewer #2 (Remarks to the Author):

Sisquella and colleagues detect genomic DNA from *P. falciparum* malaria parasites in vesicles secreted from infected red blood cells. They explore the nature and timing of this process, and describe how these vesicles activate the Stimulator of Interferon Genes (STING)-dependent signalling response in immune cells. The authors assert that this mechanism may represent a 'decoy' approach by malaria parasites that promotes their infection.

First, a disclaimer that my expertise is not in immunology or molecular biology, and I trust that other reviewers will evaluate those aspects of the work. My perspective is that the work appears thorough, and the results are exciting if a bit puzzling. My largest concern is the absence of evidence that this parasite DNA packaging in vesicles is an adaptive response of parasites to manipulate host immune response, vs. a normal functioning mechanism by which immune cells poll for intracellular pathogens in non-immune cells. The sequencing work reveals that the entire nuclear genome, as well as the mitochondrial and apicoplast organeller genomes, are packaged into vesicles, so there does not appear to be an enrichment of any packaged sequences that would be especially likely to trigger STING. In the introduction, the authors cite reference 9 (Yu et al. 2016, Immunity) to support the assertion that parasites require STING activation to grow, and

suggest therefore that STING modulation via DNA-containing vesicles is therefore a deliberate parasite strategy. Reading that paper, I do not necessarily come to the same conclusion. The Yu paper focuses on *P. yoelii*, a rodent malaria parasite, infecting an unnatural host (lab mice, rather than African tree rats). They observe a very different Type I IFN response in STING deficient mice in *P. yoelii* compared to *P. berghei*, a very divergent rodent parasite species also studied in an unnatural host (lab mice). This disparity of immune response in STING-deficient mice difficult to interpret given the unnatural host employed in these rodent parasite studies, but the very different responses raise questions as to whether and how STING manipulation by parasites would actually be adaptive. The Yu paper misleadingly refers to *P. berghei* and *P. yoelii* as 'strains' of malaria parasites, rather than species that diverged millions of years ago. The authors of the present manuscript incorrectly and misleadingly extrapolate this confusing rodent work to the strategy of human malaria parasites invading human red blood cells in line 83. This manuscript would benefit greatly from further articulation of how a presumed STING 'decoy' strategy would work in the case of human malaria parasites, or consideration of whether it would be more parsimonious to study intracellular parasite DNA packaging in exosomes as a normal function of host immunity.

Small comments:

Line 402: Genetic data offer no evidence of widespread horizontal gene transfer; sex between parasites is quite sufficient to spread drug resistance genes and other genetic variants under selection.

Reviewer #3 (Remarks to the Author):

In the study submitted Dr. Regev-Rudzki's group it is proposed that DNA-harboring vesicles shed by Plasmodium infected erythrocytes effectively activate monocytes to produce type I IFN, chemokines and pro-inflammatory cytokines. The study is divided in three main parts: (i) molecular characterization of vesicle contents; (ii) internalization of vesicles by monocytes; (iii) induction of type I IFN and chemokines mRNA expression and protein synthesis by monocytes activated with vesicles derived from parasite-infected erythrocytes. While the first part of the paper is explored into a great extent; second and third part are touched only superficially and lessen the impact of the study.

Specific comments:

1) From Figures 1 to 4 it is performed an extensive characterization of the vesicles shed by infected RBCs. A thorough analysis of their nucleic acid content and protection of parasite DNA in these vesicles is performed. In addition, they demonstrate the presence of histones and that DNA loading in these vesicles occur during the ring, but in RBCs containing trophozoites. I think this is the strong part of the study.

2) The next part of the study deals with the internalization of vesicles by monocytes. This part is not conclusive. A main flaw in this approach relates to the way the track vesicles uptake. In the first part of the study (Fig. 3) it is shown the vesicles shed by uRBCs contain way less RNA than vesicles derived from iRBCs. Then in Figure 5 they use RNA stained vesicles to track internalization of vesicles from different source. The fact RNA content of different vesicles is different makes the experiment of difficult interpretation.

3) There is no competition assay to say that vesicles from iRBCs are preferentially taken by monocytes. For example, "preferential" internalization may be a result of monocyte activation by vesicles from iRBCs and consequent enhancement of phagocytosis.

4) Despite of the strong statement, no experiments are performed to tell that vesicles go to the

monocyte cytosol. In the presented images, vesicles could be inside phagolysosomes, if they are internalized by phagocytosis. If they have been internalized by phagocytosis there is no demonstration that their content translocate to the cytosol. Do vesicles from IRBCs merge with monocyte cell surface membrane and deliver their content to the host cell cytosol? If so, this needs to be demonstrated. Then, why they are preferentially up-taken by monocytes?

5) THP1 are weird cell lines. It would be better that the experiments presented in Figure 5 are performed with primary human monocytes. The results presented in Fig S8 are very poorly presented. Image of those experiments should be presented in Fig. 5.

6) While type I IFNs are classically induced by STING path, expression of the other chemokines studied here are dependent on NFkB (RANTES) and secondary to IFN response (IP10). Are the RANTES and IP10 responses are a direct effect of STING activation or secondary to type I IFN response. Are there other receptors and signaling path being activated by vesicles shed by iRBCs? Again, these results need to be confirmed with primary human monocytes. Results with latter cells would be more meaningful.

7) Most of the literature on type I IFN and malaria is ignored. A significant literature and published in high impact journals should be discussed. Just to list few of them, Sharma et al. *Immunity*, Wu et al. *PNAS*, Liehl et al. *Nat. Med.*. There is also a list of papers dealing with the question of how parasite DNA gets into host cell to activate cytosolic DNA/RNA receptors. A classical one is from Parroche et al., *PNAS*. None of these papers dealing with this important topic are mentioned in this study.

8) A final point is a teleological one. While the authors believe that parasite are smart and activate innate immune receptors to evade host immune response, this may or may not be the case. It looks like malaria parasites activate multiple innate receptors, some have protective effect and other may regulate host deleterious immune response. For instance, different studies (Haque et al. and Sharma et al.) suggest that Type I responses protects the host from deleterious immune response. Activation of innate immune responses may result in escape from host immune response, but this needs to be confirmed.

Reviewers' comments:

Reviewer #1:

Major

1.1) The claim that the EVs containing pf-DNA are exosomes is a difficult one to make, and I'm not sure if this is crucial. The transfer itself of functional material is more important. If the authors wish to claim the EVs from uninfected RBCs are 'vesicles' and the EVs from infected RBC are 'exosomes' in the sense that they are derived from MVBs (not by protein expression and physical characteristics), then interference of the MVB-exosomes pathway, i.e. GW-treatment, siRNA against snMase or ESCRTs etc. will be necessary.

Author response: We completely agree with the reviewer's comment that the transfer of functional material is the most important aspect of the vesicles, and that their exact nature (exosomes versus another type of vesicle) is not crucial for this study.

Since these are intracellular parasites that live inside red blood cells, our initial aim in this part of the work was to strengthen the notion that the vesicles are derived from the intracellular milieu rather than the outcome of shedding of the RBC membrane. Or in other words, that the driving force for these vesicles' production is the infection by the parasites.

The reviewer is right that the best way to prove that the production of the vesicles is via MVB is to use genetic tools, such as siRNA, though, unfortunately, there are currently no such tools that can be applied to the study of the human parasite *Pf*.

We accept the reviewer's comment and have decided to:

- 1) Rephrase the first section of the results chapter so as to focus mainly on the biophysical characterization of the vesicles.
- 2) Remove figure 1B to avoid any further confusion.
- 3) Change the term used to describe the vesicles to "extracellular vesicles" (instead of 'exosomes') throughout the manuscript and figures.

1.2) The authors frequently use the term 'infected exosomes', this is confusing as they are not infectious.

Author response: We agree and have corrected this.

While the fish data of DNA in the exosomes is compelling, to understand what the authors are measuring in the bulk EVs with their omics approaches, any idea of the stoichiometry of DNA per EV/exosomes would be of relevance. Would it be possible to provide such information, for example by limiting dilution of the EVs and measuring DNA copy nrs by PCR? Alternatively, the authors could measure the concentration of EVs in their preps with particle analysis (nanosight and or TRPS).

Author response: The reviewer raises a valid point; however intrinsic complications of this biological system make it difficult to accurately address it. Namely, according to our qNano particle analysis data (Figure 1), the number of EVs per mL generated by uRBCs and iRBCs is in the same order of magnitude ($1.8E+09$ for uRBCs and $1.3E+09$ for iRBCs). However, not all the EVs released by an iRBC culture contain parasite genomic DNA. This is because in a parasite culture we need to grow iRBCs together with uRBCs (normally below 6-10 percent parasitaemia for a viable culture). For this reason, the pool of EVs from an iRBC culture contains also host secreted microvesicles (devoid of *Pf* DNA). Unfortunately, there is currently no reliable technique for separating the two types of vesicles and accurately estimating the contribution of infected RBCs vs. un-infected RBC-derived vesicles. Even if we measure the DNA copy numbers by PCR and consider the particle concentration measured by particle analysis, the stoichiometry of DNA per EV would be inaccurate. For instance, for the DNA sequencing analysis, we gained approximately 3,000ng of DNA from approximately $1.3E+09$ EV/mL, as measured by qNano particle analysis. However, any idea of stoichiometry from these values could be misleading given the abovementioned limitations.

1.3) The finding that DNA packaging and delivery via exosomes happen only during the early stages following invasion, suggests the existence of a time-dependent mechanism but would there no such differences to be expected at the m(i)RNA level?

Author response: This is an important question raised by the reviewer. In contrast to the secretion of the DNA cargo, the release of RNA molecules and proteins via *Pf*-derived EVs happens throughout the 48h blood stage cycle (at each of the 12 hours intervals). Since vesicles derived from the Trophozoite stage do contain RNA but do not induce type I IFN response (Figures 5 and 6), we concentrated on the effect of the DNA-harboring vesicles (*Pf* ring-stage-derived EVs). Other reports previously showed that microbe-RNA cargo secreted by EVs can

play an important role in cell communication. Future extensive work should be done to identify the differences in the EV-RNA profiles in each of the parasite developmental blood stages (Ring, Trophozoite and Schizont) and to determine the influence of the RNA cargo has on the host.

1.4) To increase physiological relevance; would plasma EVs from infected individuals be expected to harbor parasite DNA and be useful as a diagnostic?

Author response: One of the central aspects of the EV field is indeed the use of EVs as a diagnostic tool in diseases in which early diagnosis is a major challenge. However, in most cases of malaria infection (especially in endemic areas), the parasites can be detected simply from patient blood smears. Extracting EVs and detecting their DNA cargo will be much more difficult than using blood smears and less useful for the majority of the malaria infection cases. In the cases in which the parasitemia levels are very low, one can perform qPCR or PCR from whole blood samples to detect any remains of parasite cellular DNA or RNA, which is also an easier method than harvesting EVs first.

1.5) (minor point) The FACS data should also be presented as histograms are there some monocytes that engulf large amounts of EVs?

Author response: We agree with the reviewer and have added a new histogram Figure (Fig 5D). As can be seen from the IFC graphs, the EV uptake is normally distributed between the recipient cells – some monocytes internalize more vesicles while other less, but overall there are no two distinct sub-populations or any sub-group of recipient cells that engulf more EVs as compared to others.

1.6) The authors claim that the Pf EVs are ‘preferentially’ internalized by CD14+ peripheral monocytes and THP1 cells. However I cannot see any data from CD14- cell types. In addition the internalization is unclear.

Author response: We apologize for the misleading writing in this section and we now changed ‘preferentially’ to ‘efficiently’ internalized. The current work focused on the effect of *Pf*-derived EVs on monocytes, where we compared the uptake levels between *Pf*-iRBC-derived EVs to uRBC-derived EVs (Figs 5A-E, 6A), and we do not know the relative uptake for different cell types. In addition, to strengthen our data on the effect of uRBC vs iRBC-derived EVs on monocytes, we have now added a new figure (Fig 6C) showing the result of experiments in which primary monocytes were extracted from three healthy donors and the cellular effect of the vesicles upon uptake by the primary receiving cells was determined.

1.7) It is unlikely that the RNAs measured and transferred via the Pf EVs/exosomes have no effect at all. Can the authors really exclude other mechanisms of IFN-induction via STING? For example the authors may use a TLR mimic (poly-IC) in the STING KO cells to show that the endosomal TLR3/7/8 sensing pathway is not compromised. Is there a possibility that exoDNA is transcribed into inflammatory RNAs in the recipient cells (Chiu et al., Cell 2007)? Also cGAMP is known to be incorporated into particles and is exchanged between cells (Bridgeman et al., Science 2015), are the PF-exosomes devoid of cGAMP?

Author response: Previous work on the role of PRRs in sensing Malaria PAMPs has implicated the RNA sensing RIG-like receptors, and TLRs, in type I IFN-dependent responses in certain cell types and for certain species of *Plasmodium*. For example Yu *et al* 2016; Reference number 9] show that the rodent malaria parasite, *P.yoelii*, elicits type I IFN from mouse plasmacytoid dendritic cells via TLR7 sensing of RNA, while Wu *et al* [2014; Reference number 67] showed that an *in vivo* IFN stimulated gene signature was reduced in MDA5 KO compared to WT mice infected with *P.yoelii*. They also showed that IFNbeta elicited from a mouse macrophage cell line transfected with *P.yoelii* DNA was partially dependent on MDA5, RIG-I and RNA Pol III, thus implicating the sensing pathway that the reviewer refers to whereby AT-rich DNA is transcribed by RNA Pol III into a ligand for RIG-I [Chiu *et al*, Cell 2007]. However Sharma *et al* [2011; Reference number 10] showed that sensing of DNA from *P. falciparum* in mouse macrophages was independent of TLRs, the RIG-like receptors, and RNA Pol III. There is no evidence to date of a role for TLR3 in sensing *Plasmodium* nucleic acid. Results to date which show that specific PRRs detect *Plasmodium* are somewhat dependent on the cell type being studied and the species of *Plasmodium*.

Here, we found that the response of human monocytes to *P. falciparum* ring-stage vesicles, in terms of induction of type I IFNs and chemokines, was entirely STING-dependent (Fig 7A and 7B). This excludes an effect of vesicle RNA acting via cytosolic RNA sensing by RIG-like receptors since responses to RIG-I ligand in the STING CRISPR knock-out THP-1 cells we are using have been shown to be normal [Mankan *et al*, 2014: Reference number 50].

To clarify this we have added the following text to the results (page 11): ‘we used STING knockout (KO) THP-1 cells generated by the CRISPR/Cas9 system which were shown to be impaired in their ability to respond to cytosolic DNA, while their response to RNA was intact.’

It is highly unlikely that TLR3/7/8 are involved since these PRRs do not use STING for signalling, and also function in endosomes and not in the cytosol. Further, the fact that the STING CRISPR knock-out THP-1 cells respond normally to pppRNA (the RIG-I ligand) [Mankan *et al*, 2014: Reference number 50] excludes the possibility that EvDNA transcribed into a RIG-I ligand contributes to the type I IFN response of these cells to *P. falciparum* ring-stage vesicles. Since the RBCs from which the *Pf*-EVs originate do not contain DNA sensing machinery such as cGAS that would generate cGAMP, it is very unlikely that these EVs contain cGAMP.

Minor

1.8) Exosomes are typically 100nm in size, and the diffraction limit of light is 200nm. This suggests that it is impossible to tell whether the images in Fig 3A are indeed DNA-containing exosomes.

Author response: The reviewer is right by saying that the EVs’ size falls below the light diffraction limit. In fact, raw fluorescent images of the vesicles, taken by a wide-field microscope, appear to be quite convoluted. However, by optimizing the intensity and minimizing the bleaching of the dyes, as well as using an algorithm to deconvolute the images, we can push the technique and detect particles below the limit of resolution. The software used to deconvolve the images is SoftWorx (Applied Precision, GE Healthcare), which uses a constrained iterative

deconvolution algorithm. Still, in some images we observe a diffraction pattern, indicating that we are working at the limit of detection.

We have now added a clarifying explanation in the Methods section titled 'Widefield deconvolution microscopy (WDM)'.

1.9) The statement that deep-sequencing leads to an 'unbiased' representation is given too much credit in the procedure. It is well known that every sequencing protocol and informatics analysis leads to some sort of bias.

Author response: We agree with the reviewer and have removed the word "unbiased."

1.10) I find the figures in general not very appealing, this needs more attention. There seems to be a higher degree of RNA-transfer than DNA (figure 5) is this true? If so is this not relevant to mention?

Author response: It is difficult to compare between the RNA and DNA cargo components in terms of their uptake by recipient cells. First, the fraction of the vesicles that contain RNA is much larger than the fraction of vesicles that contain DNA (Figure 2). As shown in Figure 3, uRBC-derived vesicles contain RNA but not DNA. On top of that, differences between the two assay results could be due to the efficiency of the two different fluorescent dyes (Thiazole orange vs Hoechst).

We have now partially edited and re-arranged the figures.

1.11) Yu et al. identify a cross-regulatory mechanism of two type I IFN signaling pathways in plasmacytoid DCs (pDC), which is critical for generating protective immunity against lethal malaria infection. They show that cGAS-, STING-, MAVS-, or MDA5-deficient mice were resistant to lethal YM infection compared with wild-type (WT) mice. This suggests that these molecules are required for pathogen survival and infection. However in this paper it was also shown that serum amounts of IFN- α / β in the genetically deficient mice were abolished upon depletion of pDC or ablation of Tlr7, Myd88, and Irf7. Interestingly, EVs from pathogen-infected cells that are taken up by pDCs recognize RNA (Baglio et al., PNAS 2016) may also protect against (viral) infection. Can the authors reconcile these observations with their findings? Would it be possible that Pf EVs have an alternate function in pDCs?

Author response: Baglio *et al* [2016; Reference number 66] showed that cells latently infected with EBV can trigger anti-viral immunity in neighbouring dendritic cells due to transfer of viral RNA via exosomes. We thank the reviewer for pointing out this paper, which demonstrates a similar concept to what we reveal here, namely nucleic acid sensing of pathogens at a distance via transfer of nucleic acid in extracellular vesicles. Whether such sensing benefits, or is detrimental to, the host is probably pathogen-specific. Since Yu *et al* [2016; Reference number 9] showed that removal of STING *in vivo* increases resistance of mice to *P. yoelii*, and that STING activation in pDCs downregulates TLR-dependent type I IFN responses via SOCS1 it is possible that *Pf* EVs if taken up by human pDCs would activate STING, which could either amount to an immune subversion mechanism, or to normal innate immune pathogen detection.

In the revised manuscript we discuss these issues in more depth and cite the Baglio paper in the Discussion (Page 13/14).

1.12) Figure 7 How did the authors control for equal amount of EVs (exosomes from uRBC vs EVs from iRBCs).

Author response: Equal amounts of EVs were measured by Nanosigh (NTA particle detector) and were introduced into recipient monocytes in each experiment. We have now added a clarification in the Methods section.

1.13) The fact that the EV-DNA must reach the monocyte cytosol upon uptake, suggests an endosomal/lysosomal escape mechanism. Would there be a reason to believe that Pf-exosomes have such a mechanism?

Author response: Work from other systems suggests that EV uptake by recipient cells could involve endocytosis, phagocytosis or micropinocytosis (e.g. reviewed in Mulcahy *et al*, *J. Extracell. Vesicles*; 2014). Once taken up, how exactly the contents of the exosome is delivered into the cytosol, so that the DNA can be detected by the cytosolic sensing machinery, is unclear. For instance, in a recent study (Lian *et al*, *Cell Research* 2017) the authors tried to address the question of how self-DNA-containing exosomes deliver DNA to AIM2 in the cytosol and concluded that endosomes and lysosomes are not involved but that it is an 'exosome-specific delivery route'. Thus, at this stage we cannot speculate the exact mechanisms whereby DNA from *Pf*-derived EVs taken up by a monocyte enters the cytosol.

Reviewer #2:

My largest concern is the absence of evidence that this parasite DNA packaging in vesicles is an adaptive response of parasites to manipulate host immune response, vs. a normal functioning mechanism by which immune cells poll for intracellular pathogens in non-immune cells. The sequencing work reveals that the entire nuclear genome, as well as the mitochondrial and apicoplast organeller genomes, are packaged into vesicles, so there does not appear to be an enrichment of any packaged sequences that would be especially likely to trigger STING. In the introduction, the authors cite reference 9 (Yu et al. 2016, *Immunity*) to support the assertion that parasites require STING activation to grow, and suggest therefore that STING modulation via DNA-containing vesicles is therefore a deliberate parasite strategy. Reading that paper, I do not necessarily come to the same conclusion. The Yu paper focuses on *P. yoelii*, a rodent malaria parasite, infecting an unnatural host (lab mice, rather than African tree rats). They observe a very different Type I IFN response in STING deficient mice in *P. yoelii* compared to *P. berghei*, a very divergent rodent parasite species also studied in an unnatural host (lab mice). This disparity of immune response in STING-deficient mice difficult to interpret given the unnatural host employed in these rodent parasite studies, but the very different responses raise questions as to whether and how STING manipulation by parasites would actually be adaptive. The Yu paper

misleadingly refers to *P. berghei* and *P. yoelii* as 'strains' of malaria parasites, rather than species that diverged millions of years ago. The authors of the present manuscript incorrectly and misleadingly extrapolate this confusing rodent work to the strategy of human malaria parasites invading human red blood cells in line 83. This manuscript would benefit greatly from further articulation of how a presumed STING 'decoy' strategy would work in the case of human malaria parasites, or consideration of whether it would be more parsimonious to study intracellular parasite DNA packaging in exosomes as a normal function of host immunity.

2.1) Author response: The reviewer's points are well taken. We agree that innate immune sensing of malaria DNA, leading to type I IFN activation could either be an adaptive response of parasites to manipulate host immune response, OR a normal functioning mechanism by which immune cells sense intracellular pathogens in non-immune cells. We agree that the manuscript would benefit from further articulation and more detailed discussion of these possibilities, and we have adjusted and added to the revised manuscript accordingly.

We were careful not to exclude either the 'decoy' or 'detection' possibility in our paper, but to focus on the mechanism whereby malaria DNA could be detected by STING, since the STING DNA sensing pathway, which operates in the cytosol of immune cells such as monocytes, has been linked to responses to malaria both *in vitro* and *in vivo*. We have revised our manuscript to clarify further that such DNA responses could either be 'decoy' or 'detection', to discuss the Yu *et al* study more fully, and to put it in the context of the different species of parasites used. However, the STING literature to date does seem to suggest that in most cases of host-pathogen interactions that do not involve viruses, STING activation favours the pathogen, or certainly is detrimental to the host. For example, *P.yoelii* [Yu *et al*, 2016; Reference number 9], *Listeria monocytogenes* [Archer *et al* 2014 PLoS Path], and *M. tuberculosis* [Manzanillo *et al* 2012 Cell Host Microbe 11:469].

To address the reviewers comments, specific changes have been made to the abstract, the Introduction (page 3) and the Results section (text deleted from the top of page 10), while the Discussion has been re-structured and expanded (pages 13-14).

The reviewer also comments that 'The sequencing work reveals that the entire nuclear genome, as well as the mitochondrial and apicoplast organeller genomes, are packaged into vesicles, so there does not appear to be an enrichment of any packaged sequences that would be especially likely to trigger STING'. However STING-dependent DNA sensing, mediated by upstream DNA sensors such as cGAS and IFI16 is known to be independent of any particular DNA sequence, and simply responds to dsDNA delivered to the cytosol, since the interactions of the sensor with the DNA are all with the phosphate backbone of the DNA and not with individual bases.

Small comments:

2.2) Line 402: Genetic data offer no evidence of widespread horizontal gene transfer; sex between parasites is quite sufficient to spread drug resistance genes and other genetic variants under selection.

Author response: The reviewer is right; we have no data to support the occurrence of a HGT mechanism between the parasites. If such an event were to happen, it would be extremely rare with lower efficiency as compared to the sexual mechanism. We have rephrased the sentence accordingly.

Reviewer #3:

Specific comments:

3.1) From Figures 1 to 4 it is performed an extensive characterization of the vesicles shed by infected RBCs. A thorough analysis of their nucleic acid content and protection of parasite DNA in these vesicles is performed. In addition, they demonstrate the presence of histones and that DNA loading in these vesicles occur during the ring, but in RBCs containing trophozoites. I think this is the strong part of the study.

3.2) The next part of the study deals with the internalization of vesicles by monocytes. This part is not conclusive. A main flaw in this approach relates to the way the track vesicles uptake. In the first part of the study (Fig. 3) it is shown the vesicles shed by uRBCs contain way less RNA than vesicles derived from iRBCs. Then in Figure 5 they use RNA stained vesicles to track internalization of vesicles from different source. The fact RNA content of different vesicles is different makes the experiment of difficult interpretation.

Author response: The reviewer raises a very important comment. Despite using equal amounts of vesicles (derived from *Pf*-iRBC and from uRBCs) and pre-measuring their RNA fluorescent intensity before treating recipient cells (using a NTA system), as the reviewer points out, the fact the RNA-cargo quantities are different between the two types of vesicles may bias the experimental conclusion.

To overcome this matter, we have now added an entire Figure (Fig S8 A-B), in which we have optimized the assay to use a lipid-based stain (rather than RNA-based). The lipid dye (Dil; Thermo Fisher Scientific) stains the lipids of the vesicle membrane. In this case, there is no bias towards the specific content of each vesicle type. As seen in Figure S8, and in accordance to our previous data, uptake of the vesicles following Dil staining demonstrates greater intake by vesicles derived from *Pf*-iRBC as compared to vesicles derived from uRBCs.

3.3) There is no competition assay to say that vesicles from iRBCs are preferentially taken by monocytes. For example, “preferential” internalization may be a result of monocyte activation by vesicles from iRBCs and consequent enhancement of phagocytosis.

Author response: We apologize for the misleading writing in this section and we now changed ‘preferentially’ to ‘efficiently’ internalized. The current work focuses on the effect of *Pf*-derived EVs on monocytes, where we compared the uptake levels between *Pf*-iRBC-derived EVs to uRBC-derived EVs (Figs 5A-E 6A), and we do not know the relative uptake for different cell types.

3.4) Despite of the strong statement, no experiments are performed to tell that vesicles go to the monocyte cytosol. In the presented images, vesicles could be inside phagolysosomes, if they are internalized by phagocytosis. If they have been internalized by phagocytosis there is no demonstration that their content translocate to the cytosol. Do vesicles from IRBCs merge with monocyte cell surface membrane and deliver their content to the host cell cytosol? If so, this needs to be demonstrated. Then, why they are preferentially up-taken by monocytes?

Author response: We understand the reviewer's question; the vesicle uptake mechanism still remains mostly unsolved in many EV systems and will require extensive future work to tackle the exact mechanism of the cargo release. However, by performing additional kinetic measurements using IFC, we now demonstrate that the final destination of the DNA cargo in the immune cell progressively expands within the cytosol area over the course of 45 minutes rather than is concentrated within distinct regions (e.g., phagolysosomes) (New Fig S9 A-C). To generate these data by IFC, we scanned at least 50,000 recipient immune cells and fluorescently tracked the cargo destination over time in live cells. As seen in Fig S9, within the first 15 minutes, the EV Hoechst signal appeared as clear bright spots inside the cells, followed by the signal dispersing around the cell. To quantify the cellular distribution of the EVs, we calculated the area of the highest intensity pixels (Area_Threshold) - small bright spots have a low area, and the dye's even dispersion throughout the cell corresponds to a higher area. The results indeed demonstrate an increase in the area of the highest intensity pixels, in accordance with the distribution of the Hoechst signal within the cell (S9A-B). More specifically, the upper graph (9A) represents the area of the highest intensity stained pixels (Area_Threshold) of the HO staining along time (up to 45 minutes of live EV uptake measurement). The bottom graph (B) represents the HO intensity detected over time, demonstrating the HO accumulation within the cell area. Please see response to point 1.13 above also.

3.5) THP1 are weird cell lines. It would be better that the experiments presented in Figure 5 are performed with primary human monocytes. The results presented in Fig S8 are very poorly presented. Image of those experiments should be presented in Fig. 5.

Author response: we have now added a new figure (Fig 6C), in which we measure RNA expression from each of the genes in question (*IFN α* , *IFN β* , *RANTES*, *IFIT* and *IP10*) using human primary monocytes that we extracted from three healthy donors. These results support our data in THP-1 cells and confirm the induction of the response.

Although the use of every cell line comes with provisos, for cytosolic DNA sensing studies involving both isolated DNA and pathogens THP-1 cells have been well validated and shown to recapitulate the DNA sensing pathways in primary human monocytes. For example, we have shown that the type I IFN and IP10 response of THP-1 cells and primary human CD14+ monocytes to dsDNA is similarly sensitive to siRNA targeting IFI16, cGAS and STING (unpublished data). The ability to generate CRISPR/Cas9 KO THP-1 cells lacking STING makes their use advantageous over other currently available human monocytic cell line models.

3.6) While type I IFNs are classically induced by STING path, expression of the other chemokines studied here are dependent on NFkB (RANTES) and secondary to IFN response (IP10). Are the RANTES and IP10 responses are a direct effect of STING activation or secondary to type I IFN response. Are there other receptors and signaling path being activated by vesicles shed by

iRBCs? Again, these results need to be confirmed with primary human monocytes. Results with latter cells would be more meaningful.

Author response: The mechanism whereby the cGAS-STING sensing pathway activates IRF3 via TBK1 has been well characterised, and provides an explanation as to how cytosolic DNA induces IRF3-dependent genes such as IFN β via STING. However, although the mechanisms are less well described, we and others have consistently shown that cytosolic DNA-dependent activation of NF κ B-dependent genes are also STING-dependent. Both IP10 and RANTES induction are both NF κ B- and IRF3-dependent, and inhibition of either transcription factor inhibits gene induction. Thus we consistently find that DNA-stimulated RANTES and IP10 are STING-dependent, and that they are primary response genes (and not dependent on type I IFN signalling), since they are induced by the primary response pathways activated by STING (IRF3 and NF κ B).

As regards the potential role of other signalling pathways, the genes measured here are shown to be completely STING-dependent (Fig 7A). In terms of the role of other PRRs in sensing malaria nucleic acid, see response to point 1.7 above.

3.7) Most of the literature on type I IFN and malaria is ignored. A significant literature and published in high impact journals should be discussed. Just to list few of them, Sharma et al. Immunity, Wu et al. PNAS, Liehl et al. Nat. Med.. There is also a list of papers dealing with the question of how parasite DNA gets into host cell to activate cytosolic DNA/RNA receptors. A classical one is from Parroche et al., PNAS. None of these papers dealing with this important topic are mentioned in this study.

Author response: We apologize for the omissions, and have included all the papers mentioned by the reviewer, and others on malaria and IFN, in a revised and expanded Discussion, pages 13-14. We tried to focus the original paper submission specifically on STING for clarity, since type I IFN induction is only one outcome of STING-dependent sensing (albeit a very important and well defined outcome), and since the literature on type I IFN and malaria is complex and varies according to the particular plasmodium species examined.

3.8) A final point is a teleological one. While the authors believe that parasites are smart and activate innate immune receptors to evade host immune response, this may or may not be the case. It looks like malaria parasites activate multiple innate receptors, some have protective effect and other may regulate host deleterious immune response. For instance, different studies (Haque et al. and Sharma et al.) suggest that Type I responses protect the host from deleterious immune response. Activation of innate immune responses may result in escape from host immune response, but this needs to be confirmed.

Author response: The reviewer's point is well taken. We have adjusted the revised manuscript to address this teleological issue – please see response to point 2.1 above.

REVIEWERS' COMMENTS:

Reviewer #1 (Remarks to the Author):

I have read the rebuttal of the authors to the reviewers questions and find that the concerns were adequately addressed with a few minor points remaining. Overall this is an interesting study that will advance the field.

- I agree that stoichiometry of nucleic acids associated with vesicles remains difficult to assess due to numerous (technical) limitations. However this remains an important, if not, critical issue concerning the physiological relevance of these EVs. I recommend that the authors make the reader aware of this issue in the ms.

- The authors argue that STING-mediated IFN activation is the only pathogen sensing mechanism of EVs loaded with Pf-DNA. Their data is indeed consistent with this. However, I would argue that transfection of purified EV-RNAs from infected RBC, is also likely to activate IFNs, possible via TLR7 (Baccarella et al., Inf. Imm. 2013). Since these experiments were not performed the authors cannot exclude that such a mechanism functions in parallel, possibly with different kinetics and depending on the recipient cell.

- references should be checked for adequacy.

- The labeling of the figures (6-7) should be improved

Reviewer #3 (Remarks to the Author):

Authors have appropriately addressed by comments and suggestions.

Point-by-point response to final issues raised by referees:

Reviewer #1 (Remarks to the Author):

I have read the rebuttal of the authors to the reviewers questions and find that the concerns were adequately addressed with a few minor points remaining. Overall this is an interesting study that will advance the field.

We thank the reviewer for these comments.

- I agree that stoichiometry of nucleic acids associated with vesicles remains difficult to assess due to numerous (technical) limitations. However this remains an important, if not, critical issue concerning the physiological relevance of these EVs. I recommend that the authors make the reader aware of this issue in the ms.

A comment to the reader was added to the Discussion section (page 13).

- The authors argue that STING-mediated IFN activation is the only pathogen sensing mechanism of EVs loaded with Pf-DNA. Their data is indeed consistent with this. However, I would argue that transfection of purified EV-RNAs from infected RBC, is also likely to activate IFNs, possible via TLR7 (Baccarella et al., Inf. Imm. 2013). Since these experiments were not performed the authors cannot exclude that such a mechanism functions in parallel, possibly with different kinetics and depending on the recipient cell.

We have added text to the Discussion (page 14), with the reference, to include this possibility.

- references should be checked for adequacy.

References have been checked

- The labeling of the figures (6-7) should be improved

This comment is not entirely clear

Many thanks,

Andrew Bowie and Neta Regev-Rudzki